# Ribosome subunit attrition and activation of the p53–MDM4 axis dominate the response of MLL-rearranged cancer cells to WDR5 WIN site inhibition

Gregory Caleb Howard[1], Jing Wang[2,3], Kristie L Rose[4,5], Camden Jones[1], Purvi Patel[4], Tina Tsui[4], Andrea C Florian[1†], Logan Vlach[6], Shelly L Lorey[1], Brian C Grieb[6], Brianna N Smith[6], Macey J Slota[1‡], Elizabeth M Reynolds[1], Soumita Goswami[1], Michael R Savona[6], Frank M Mason[6], Taekyu Lee[5], Stephen Fesik[5,7,8], Qi Liu[2,3], William P Tansey[1,5]*

[1]Department of Cell and Developmental Biology, Vanderbilt University School of Medicine, Nashville, United States; [2]Department of Biostatistics, Vanderbilt University Medical Center, Nashville, United States; [3]Center for Quantitative Sciences, Vanderbilt University Medical Center, Nashville, United States; [4]Mass Spectrometry Research Center, Vanderbilt University School of Medicine, Nashville, United States; [5]Department of Biochemistry, Vanderbilt University School of Medicine, Nashville, United States; [6]Department of Medicine, Vanderbilt University Medical Center, Nashville, United States; [7]Department of Pharmacology, Vanderbilt University School of Medicine, Nashville, United States; [8]Department of Chemistry, Vanderbilt University, Nashville, United States

*For correspondence:
william.p.tansey@vanderbilt.edu

Present address: †Department of Biology, Belmont University, Nashville, United States; ‡Department of Urology, University of California San Francisco, San Francisco, United States

**Abstract** The chromatin-associated protein WD Repeat Domain 5 (WDR5) is a promising target for cancer drug discovery, with most efforts blocking an arginine-binding cavity on the protein called the 'WIN' site that tethers WDR5 to chromatin. WIN site inhibitors (WINi) are active against multiple cancer cell types in vitro, the most notable of which are those derived from MLL-rearranged (MLLr) leukemias. Peptidomimetic WINi were originally proposed to inhibit MLLr cells via dysregulation of genes connected to hematopoietic stem cell expansion. Our discovery and interrogation of small-molecule WINi, however, revealed that they act in MLLr cell lines to suppress ribosome protein gene (RPG) transcription, induce nucleolar stress, and activate p53. Because there is no precedent for an anticancer strategy that specifically targets RPG expression, we took an integrated multi-omics approach to further interrogate the mechanism of action of WINi in human MLLr cancer cells. We show that WINi induce depletion of the stock of ribosomes, accompanied by a broad yet modest translational choke and changes in alternative mRNA splicing that inactivate the p53 antagonist MDM4. We also show that WINi are synergistic with agents including venetoclax and BET-bromodomain inhibitors. Together, these studies reinforce the concept that WINi are a novel type of ribosome-directed anticancer therapy and provide a resource to support their clinical implementation in MLLr leukemias and other malignancies.

## eLife assessment

This **important** article reveals that one of the major roles of the WDR5 WIN site is to promote ribosome synthesis, and that by attacking the WIN site with inhibitors ribosome attrition occurs creating new vulnerabilities that can be therapeutically exploited. This deficiency of ribosomal proteins also

provokes the p53 response. The data from a variety of approaches is generally very **convincing**, and together buttresses the authors' conclusions and interpretations quite nicely; overall, this article will provide a justification for pre-clinical and translational studies of WDR5 interaction site inhibitors.

## Introduction

WDR5 is a highly conserved protein that moonlights in a variety of functions in the nucleus. It rose to prominence as a component of epigenetic writer complexes that deposit histone H3 lysine 4 (H3K4) methylation (*Guarnaccia and Tansey, 2018*), but was subsequently found to act outside these complexes to facilitate the integrity of the mitotic spindle (*Ali et al., 2017*), bookmark genes for reactivation after mitosis (*Oh et al., 2020*), and promote transcription of a subset of ribosomal protein genes [RPGs]; (*Bryan et al., 2020*) via recruitment of the oncoprotein transcription factor MYC to chromatin (*Thomas et al., 2019*). WDR5 is also frequently overexpressed in cancer, where its over-expression correlates with aggressive disease and poor clinical outcomes (*Guarnaccia and Tansey, 2018*). Accordingly, WDR5 is an auspicious target for inhibition in a range of malignancies including MLL-rearranged (MLLr) leukemias (*Cao et al., 2014*; *Aho et al., 2019a*), MYC-driven cancers (*Aho et al., 2019b*), C/EBPα-mutant leukemias (*Grebien et al., 2015*), p53 gain-of-function cancers (*Zhu et al., 2015*), neuroblastomas (*Bryan et al., 2020*), rhabdoid tumors (*Florian et al., 2022*), and meta-static breast cancers (*Cai et al., 2022*).

Although WDR5 PROTACs have been described (*Yu et al., 2021*; *Li et al., 2022*; *Yu et al., 2023*), safety concerns over destroying a pan-essential protein such as WDR5 (*Siladi et al., 2022*) means that most drug discovery efforts have focused on small-molecule inhibition of key binding sites on the protein. Some initiatives target a hydrophobic cleft on WDR5 known as the 'WDR5-binding motif' (WBM) site (*Macdonald et al., 2019*; *Chacón Simon et al., 2020*) that contacts MYC (*Thomas et al., 2015*). But the majority of efforts target the 'WDR5-interaction' (WIN) site of WDR5 (*Guarnaccia and Tansey, 2018*)—an arginine binding cavity that tethers WDR5 to chromatin (*Aho et al., 2019a*) and makes contact with partner proteins carrying an arginine-containing 'WIN' motif (*Guarnaccia et al., 2021*). Multiple WIN site inhibitors (WINi) have been described (*Bolshan et al., 2013*; *Karatas et al., 2013*; *Senisterra et al., 2013*; *Cao et al., 2014*; *Grebien et al., 2015*; *Li et al., 2016*; *Karatas et al., 2017*; *Wang et al., 2018*; *Aho et al., 2019a*; *Tian et al., 2020*; *Chen et al., 2021a*; *Chen et al., 2021b*), including those that are orally bioavailable and have antitumor activity in vivo (*Chen et al., 2021b*; *Teuscher et al., 2023*). Given the intense interest in developing WINi for cancer therapy, and the rapid pace of improvement in these molecules, it is likely that WINi will be ready for clinical vetting in the near future.

That said, controversy remains regarding the mechanism of action of WINi, even in the context of MLLr leukemias, where there is strong empirical support for their utility (*Weissmiller et al., 2024*). MLL-rearranged leukemias are defined by translocation of one copy of *MLL1*—a gene that encodes one of six MLL/SET proteins that are the catalytic subunits of the histone methyltransferase (HMT) complexes responsible for H3K4 methylation (*Guarnaccia and Tansey, 2018*). The near universal retention of a pristine copy of *MLL1* in these cancers led to the idea that MLLr leukemias depend on wild-type MLL1 to support the activity of oncogenic MLL1-fusion oncoproteins (*Thiel et al., 2010*)—a function in turn that depends on insertion of a low-affinity WIN motif within MLL1 into the WIN site of WDR5 (*Alicea-Velázquez et al., 2016*). Consistent with this notion, early peptidomimetic WINi are active against MLLr leukemia cells in vitro and are reported to suppress levels of H3K4 methylation at canonical MLL1-fusion target genes such as the *HOXA* loci, causing cellular inhibition through a combination of differentiation and apoptosis (*Cao et al., 2014*). Subsequently, however, wild-type MLL1 was shown to be dispensable for transformation by MLL-fusion oncoproteins (*Chen et al., 2017*), and our analysis of picomolar small-molecule WINi revealed that they act in MLLr cells without inducing significant changes in the expression of *HOXA* genes or levels of H3K4 methylation (*Aho et al., 2019a*). Instead, WINi displace WDR5 from chromatin and directly suppress the transcription of ~50 genes, the majority of which are connected to protein synthesis, including half the cohort of RPGs. We also found that WINi provoke nucleolar stress and induce p53-dependent cell death. Based on our findings, we proposed that WINi kill MLLr cells via depletion of part of the ribosome inventory that induces apoptosis via a ribosome biogenesis stress response.

The concept of ribosome-directed cancer therapies is not new (*Laham-Karam et al., 2020*; *Temaj et al., 2022*). Besides mTOR and translational inhibitors, one of the most prevalent strategies in this realm is inhibition of ribosomal RNA (rRNA) production or processing, which is a feature of both existing chemotherapies such as platinum-containing compounds (*Bruno et al., 2017*), as well as newer targeted RNA polymerase I inhibitors (*Drygin et al., 2011*; *Peltonen et al., 2014*). Although these agents exert their anticancer effects through multiple mechanisms (*Laham-Karam et al., 2020*), they are generally thought to disrupt the stoichiometry of RNA and protein components of the ribosome, leading to an excess of ribosomal proteins that inactivate MDM2 to induce p53-dependent cancer cell death. The paradigm we developed for WINi is modeled after that of rRNA inhibitors, although it is important to note that a significant point of divergence from rRNA poisons is that in this model WINi induce p53 not by promoting excess ribosomal protein accumulation, but by causing a selective imbalance in the ribosome subunit inventory. How such an imbalance could lead to p53 induction, as well as other consequences it may have on cellular processes, remains unknown.

Fortifying understanding of the mechanism of action of WINi in MLLr cancer cells is key to their clinical implementation. At present, there is no precedent for the mechanism we propose, no understanding of the impact of selective ribosome subunit depletion on translation or other tumor-relevant processes, and no expectations for how resistance to WINi could emerge or how their antitumor actions could be made more effective. To ameliorate these deficiencies, we took an integrated multi-omics approach, combining transcriptional and translational profiling with genome-wide CRISPR screening to probe WINi action in MLLr cells. Our studies show that although the transcriptional effects of WINi on ribosome subunit expression are confined to those RPGs directly regulated by WDR5, effects at the protein level are not, and WIN site inhibition leads to diminution of the entire stock of cytosolic ribosomes. Ribosome subunit attrition is accompanied by a broad translational choke, induction of nucleolar stress, and activation of p53—driven in large part via RPL22-dependent alternative splicing of the p53 antagonist MDM4. We also show that WINi are synergistic with approved and targeted agents including venetoclax and BET-bromodomain inhibitors. Collectively, these findings solidify a novel mechanism of action for WINi in MLLr cells and highlight a path for their optimal clinical implementation.

## Results

### Impact of WINi on the transcriptome of MLLr cancer cells

Our model for the action of WINi in MLLr leukemia cells is based on analysis of two early-generation compounds (*Aho et al., 2019a*): C3 ($K_d$ = 1.3 nM) and C6 ($K_d$ = 100 pM). Subsequently (*Tian et al., 2020*), we discovered more potent molecules such as C16 ($K_d$ < 20 pM) that have not been extensively profiled. To determine if improvements in the potency of WINi have resulted in divergent activities, we first compared C6 with C16 (*Figure 1A*). Both molecules bind the WIN site of WDR5 (*Figure 1B*, *Figure 1—figure supplement 1A*), but differ in affinity due to a bicyclic dihydroisoquinolinone core that locks C16 into a favorable binding conformation (*Figure 1C*). Consistent with its higher affinity, C16 is ~20 times more potent than C6 in inhibiting MV4;11 (MLL–AF4) and MOLM13 (MLL–AF9) leukemia lines (*Figure 1—figure supplement 1B and C*): in MV4;11 cells, for example, the $GI_{50}$ for C6 is 1 µM compared to 46 nM for C16. These differences in potency are reflected at the level of RPG suppression. Using a target engagement assay (*Florian et al., 2022*) that measures transcript levels from seven RPGs—five (*RPS14*, *RPS24*, *RPL26*, *RPL32*, and *RPL35*) that are always bound by WDR5 and two (*RPS11* and *RPS14*) that are never bound—we observe that maximal suppression of RPG transcripts occurs at ~2 µM for C6 and ~100 nM for C16 in MV4;11 (*Figure 1D*) and MOLM13 (*Figure 1—figure supplement 1D*) cells. To functionally compare these two inhibitors of different potencies, we used these RPG-normalized doses in all our subsequent studies.

We performed RNA-sequencing (RNA-seq) on MV4;11 cells treated for 48 hr with DMSO, C6, or C16 (*Figure 1—figure supplement 1E*, *Figure 1—source data 1*). Spike-in controls were not included. Both compounds elicit thousands of gene expression changes (*Figure 1E*), a majority of which are less than twofold in magnitude (*Figure 1—figure supplement 1F*). We had previously performed RNA-seq on MV4;11 cells treated with 2 µM C6 and observed just ~75 induced and ~460 reduced transcripts (*Aho et al., 2019a*). In this earlier work, however, increased variance among replicates made it difficult for as many small gene expression changes to reach statistical significance

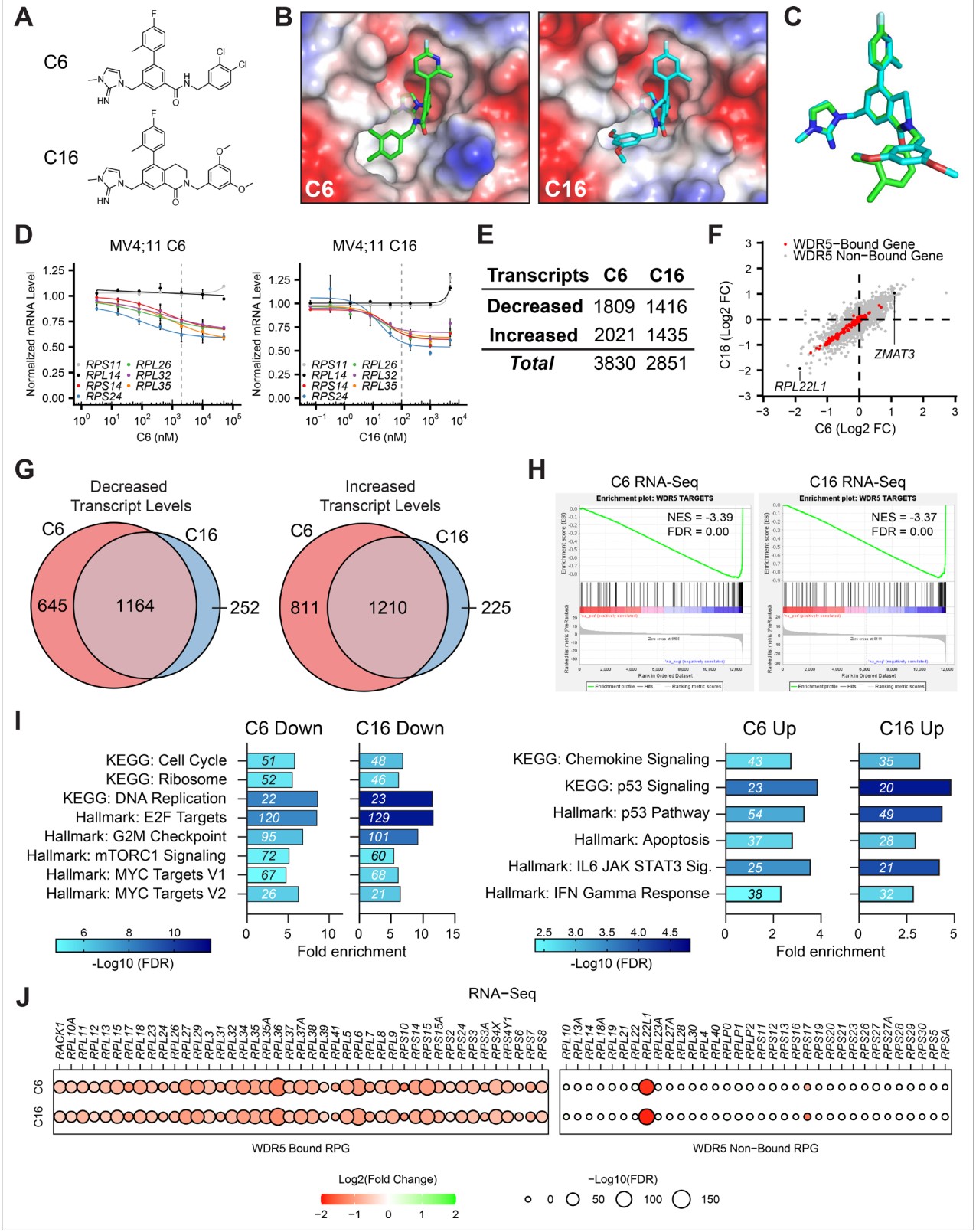

**Figure 1.** Impact of WIN site inhibitors (WINi) on the transcriptome of MLLr cancer cells. (**A**) Chemical structures of C6 and C16. (**B**) Crystal structures of C6 or C16 bound to the WIN site of WDR5 with electrostatic surfaces mapped (PDB IDs: 6E23 [**Aho et al., 2019a**]; 6UCS [**Tian et al., 2020**]). The image shows a close-up view of the WIN site. (**C**) Superimposed WIN site-binding conformations of C6 (green) and C16 (blue). (**D**) Transcript levels as determined by QuantiGene analysis of representative WDR5-bound (color) or non-bound (grayscale) ribosomal protein genes in MV4;11 cells treated

*Figure 1 continued on next page*

*Figure 1 continued*

with a serial dilution range of either C6 (left) or C16 (right) and relative to DMSO-treated cells (n = 2–3; mean ± SEM). Vertical dashed line indicates either 2 µM C6 (left) or 100 nM C16 (right). (**E**) Number of genes with significantly (false discovery rate [FDR] < 0.05) altered transcript levels following treatment of MV4;11 cells with C6 (2 µM) or C16 (100 nM) for 48 hr, as determined by RNA-Seq (n = 3). See *Figure 1—source data 1* for complete output of RNA-seq analysis. (**F**) Comparison of gene expression changes elicited by C6 (x-axis) and C16 (y-axis), represented as Log2 fold change (FC) compared to DMSO. WDR5-bound genes are colored red. Locations of *RPL22L1* and *ZMAT3* are indicated. (**G**) Overlap of genes with decreased (left) or increased (right) transcript levels in MV4;11 cells treated with C6 or C16. (**H**) Gene set enrichment analysis (GSEA) showing the distribution of genes suppressed in MV4;11 cells in response to C6 (left) or C16 (right) against the list of all genes bound by WDR5 in those cells (*Aho et al., 2019a*). NES, normalized enrichment score. (**I**) Enrichment analysis of genes suppressed (left) or induced (right) by C6 or C16 in MV4;11 cells. KEGG and Hallmark. MSigDB pathways are shown. Fold enrichment of indicated pathways is presented on the x-axis, the number of genes is shown in italics in each bar, and colors represent -Log10 FDR. See *Figure 1—source data 2* for additional GSEA (Hallmark) and over-representation analysis (ORA) (Hallmark) analyses of differentially expressed genes. (**J**) Transcript level changes in WDR5-bound (left) and non-bound (right) RPGs elicited by C6 (top) or C16 (bottom).

The online version of this article includes the following source data and figure supplement(s) for figure 1:

**Source data 1.** Output of RNA-seq analysis of MV4;11 cells treated with C6/C16.

**Source data 2.** GSEA Hallmark and over-representation analysis (ORA) Hallmark enrichment analysis of differentially expressed genes in RNA-seq.

**Figure supplement 1.** Transcript changes elicited by WIN site inhibitors (WINi) in MLLr cancer cells.

**Figure supplement 2.** Impact of WIN site inhibitors (WINi) on *RPL22L1* and p53 target gene expression.

as in the current study (*Figure 1—figure supplement 1G*). Comparing the new RNA-seq datasets, we observe similar effects of C6 and C16 on the MV4;11 cell transcriptome (*Figure 1F*), with more than 80% of the transcripts altered by C16 altered in the same direction by C6 (*Figure 1G*). In both cases, suppressed genes are enriched in those bound by WDR5 in MV4;11 cells (*Figure 1H*), although a majority of transcriptional changes occur at loci bereft of detectable WDR5 binding (*Figure 1F*). For both compounds, expression of genes connected to protein synthesis, the cell cycle, DNA replication, mTORC signaling, and MYC are reduced, while expression of those connected to chemokine signaling, apoptosis, and p53 is induced (*Figure 1I*, *Figure 1—source data 2*). Indeed, ~90 'consensus' p53 target genes (*Fischer, 2017*) are induced by C6/C16 (*Figure 1—figure supplement 2A and B*), including the tumor suppressor *ZMAT3* (*Figure 1F*; *Bieging-Rolett et al., 2020*). Finally, we note that for both compounds the transcriptional effects on RPG expression are almost entirely confined to those RPGs bound by WDR5 (*Figure 1J*). The conspicuous exception to this trend is *RPL22L1*—a paralog of *RPL22*—mRNA levels which are strongly reduced by C6/C16 (*Figure 1F and J*). This feature of the response is not confined to MLLr cells as *RPL22L1* expression is also decreased by WINi in sensitive rhabdoid tumor cell lines (*Florian et al., 2022*), but not in the insensitive K562 leukemia line (*Bryan et al., 2020*; *Figure 1—figure supplement 2C*). Despite not being a direct WDR5 target gene, therefore, *RPL22L1* expression is recurringly suppressed by WINi in responsive cancer cell lines.

Together, these data reveal that improvements in the potency of WINi have not resulted in substantive changes in their impact on the transcriptome of MLLr cells and reinforce the concept that inhibition of select RPG expression—and induction of a p53-related transcriptional program—defines the response of the transcriptome to WIN site blockade in this setting.

## Impact of WINi on the translatome of MLLr cancer cells

We previously showed that treatment of MV4;11 cells with C6 results in a time-dependent decrease in translation as measured by bulk labeling of nascent polypeptide chains with *O*-propargyl-puromycin (OPP) (*Aho et al., 2019a*). We confirmed this finding with C6 and extended it to C16, showing that while there is no significant effect on protein synthesis capacity after 24 hr of WINi treatment, a progressive decline begins at 48 hr, reaching an ~40% reduction at the 96 hr treatment point (*Figure 2—figure supplement 1A and B*). To determine more precisely the effects of WIN site inhibition on translational processes, we performed ribosome profiling (Ribo-seq; *McGlincy and Ingolia, 2017*) in parallel with the RNA-seq analyses described above. Spike-in controls were not included. By sequencing ribosome-protected fragments (RPFs) in 48-hr-treated and control cells, and normalizing to transcript levels from RNA-seq, we calculated the translation efficiency (TE) of each transcript and used this to determine how C6/C16 influence translation, independent of effects on mRNA abundance.

In these experiments, RPFs have the characteristic length of ribosome-protected mRNA fragments (28–32 nucleotides; *Figure 2—figure supplement 2A*), are enriched in coding sequences

(*Figure 2—figure supplement 2B*), and map to the expected reading frame (*Figure 2—figure supplement 2C*), all of which indicate successful profiling. In contrast to RNA-seq, where we see equal numbers of transcript increases and decreases in response to C6/C16, the overwhelming effect of these compounds on translation efficiency is inhibitory (*Figure 2A*). Of the ~10,000 transcripts profiled, between ~4,500 (C16) and ~5,900 (C6) transcripts show decreased TE compared to less than 10 transcripts with increased translation (*Figure 2B*, *Figure 2—source data 1*). As we observed in the RNA-seq, changes in TE are generally less than twofold (*Figure 2A*) and there is extensive overlap between the two inhibitors, with ~90% of the transcripts decreased in TE by C16 also decreased by C6 (*Figure 2C*). In general, C6/C16 reduce translation of mRNAs in a manner independent of basal translation efficiencies (*Figure 2D*), although if we bin transcripts according to basal TE we observe that the number of highly translated transcripts (fourth quartile) impacted by C6/C16 is greater than for those transcripts with lower basal TE (*Figure 2E*). Within these quartiles, however, the magnitude of reduction in TE is equivalent (*Figure 2—figure supplement 2D*). Interestingly, mRNAs carrying better matches to the 5'TOP motif—which links translation to mTORC1 signaling (*Philippe et al., 2020*)—show less decrease in TE compared to those with poorer matches (*Figure 2—figure supplement 2E*), suggesting that mTORC1-regulated mRNAs may be spared from the full translational effects of WIN site inhibition.

Interrogating specific changes in TE induced by C6/C16, we see that the biological categories of transcripts with decreased TE echo many of those observed with decreased mRNA levels, but include more genes in each category. For example, manually curating each list for transcripts encoding the ~60 validated substrates of the protein arginine methyltransferase PRMT5 (*Radzisheuskaya et al., 2019*) reveals that 42 are translationally suppressed by C6/C16 compared to just 10 that are suppressed at the mRNA level (*Figure 2—figure supplement 2F*). Probing for Hallmark categories in the Human Molecular Signatures Database (MSigDB; *Liberzon et al., 2015*) uncovers the extent of this phenomenon (*Figure 2—source data 2*), with categories linked to MYC, E2F, mTORC1 signaling, and the G2M checkpoint (*Figure 2F*) all represented by more genes in the ribosome profiling than in the RNA-seq experiments. Within these categories, a majority of genes suppressed by C6/C16 at the mRNA level are also further suppressed translationally (*Figure 2—figure supplement 2G*). This is not a general trend in the response, however, as fewer than half of the total transcripts with reduced mRNA abundance experience this additional translational inhibition (*Figure 2—figure supplement 2H*). The finding that TE changes induced by C6/C16 extend the biological characteristics of changes in mRNA abundance may indicate a role for impaired translation in contributing to at least some of the mRNA level differences triggered by WINi. Indeed, comparing transcripts with decreased TE but no mRNA decrease with transcripts with decreased TE and mRNA levels reveals the latter are enriched in so-called 'optimal codons' (*Wu et al., 2019*) that normally promote mRNA stability but are linked to mRNA instability when translation is inhibited (*Figure 2—figure supplement 2I*).

Implicit in the previous discussion, a majority of the translational decreases triggered by C6/C16 occur at transcripts for which there are no significant changes in mRNA abundance (*Figure 2—figure supplement 2H*). Expectedly, genes with decreased TE but no mRNA level changes are enriched in several of the major Hallmark categories described above (*Figure 2—source data 3*). But we also observe enrichment in genes connected to the proteasome, spliceosome, mRNA surveillance, and translation (*Figure 2G*). The latter category includes subunits of the mitochondrial ribosome (*Figure 2—figure supplement 3A*), translation and ribosome biogenesis factors, and an expanded cohort of transcripts from RPGs (*Figure 2H*). Indeed, compared to mRNA levels, where C6/C16-induced changes are confined (with the exception of *RPL22L1*) to a decrease in expression of WDR5-bound RPGs, translational effects are not, and some of the most pronounced TE changes occur at non-WDR5 ribosomal protein target genes. Thus, beyond what we have been able to infer from previous studies, WINi causes a widespread reduction in the ability of MLLr cells to efficiently translate mRNAs connected to almost every aspect of protein synthesis and homeostasis.

## Impact of WINi on the ribosome inventory of MLLr cancer cells

Based on the finding that WDR5 controls expression of half the RPGs, we speculated that WINi induce a ribosome subunit imbalance that leads to induction of p53 (*Aho et al., 2019a*). It is also possible, however, that quality control mechanisms deplete the entire inventory of ribosomal proteins (RP) during prolonged WIN site blockade. To distinguish between these possibilities, we tracked changes

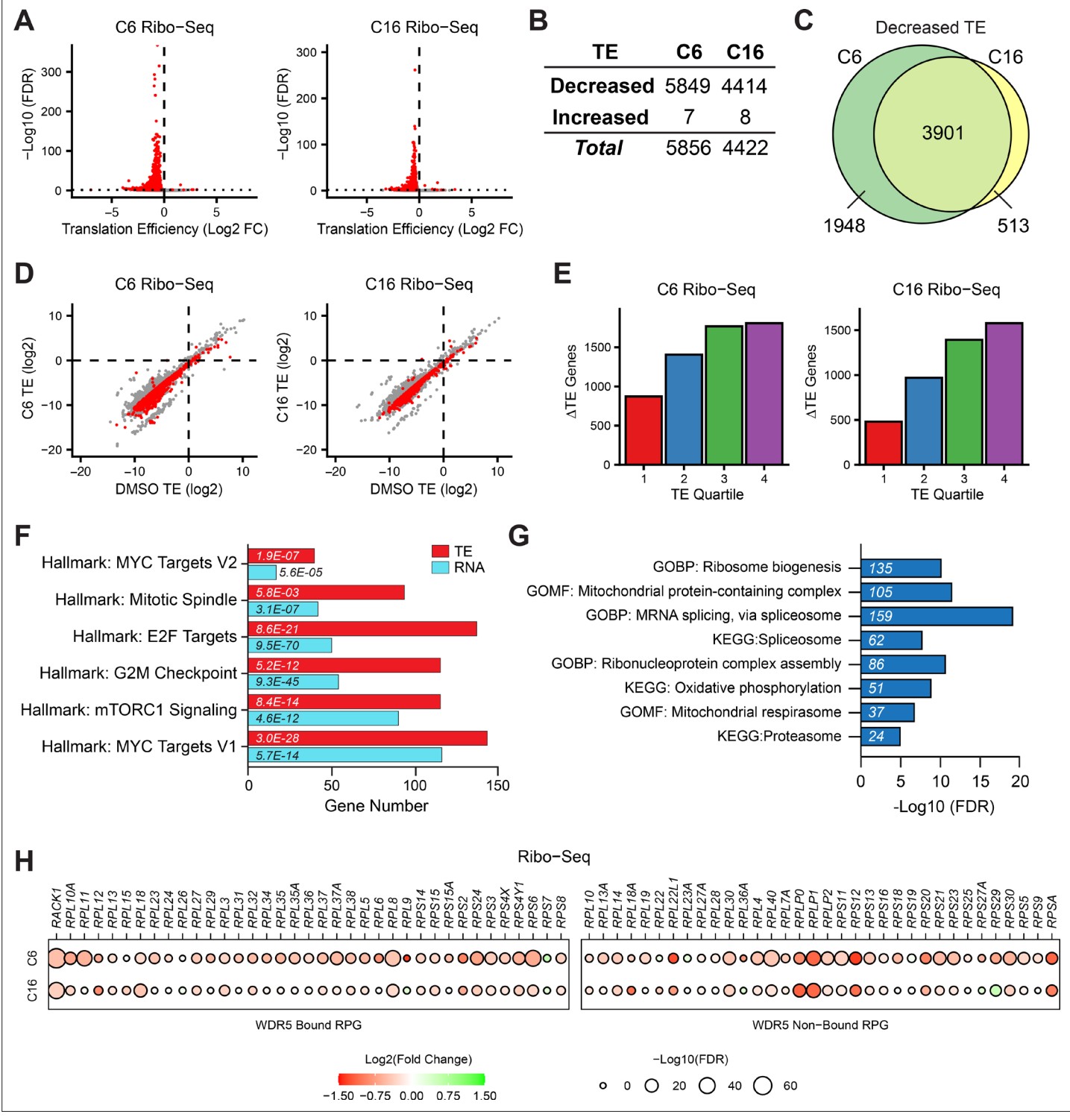

**Figure 2.** Impact of WIN site inhibitors (WINi) on the translatome of MLLr cancer cells. (**A**) Volcano plots depicting alterations in translation efficiency (TE) induced by 48 hr treatment of MV4;11 cells with either 2 μM C6 (left) or 100 nM C16 (right) compared to DMSO (n = 2; red indicates false discovery rate [FDR] < 0.05 and Log2 FC > 0.25), as determined by Ribo-seq. (**B**) Number of mRNAs with significantly (FDR < 0.05 and Log2 FC > 0.25) altered TE levels following treatment of MV4;11 cells with C6 (2 μM) or C16 (100 nM) for 48 hr. See *Figure 2—source data 1* for complete output of Ribo-seq analysis. (**C**) Overlap of mRNAs with significantly decreased TE in response to C6 or C16 treatment. (**D**) TE of mRNAs in DMSO-treated MV4;11 cells plotted against translation efficiencies of mRNAs in cells treated with either C6 (left) or C16 (right). Red indicates mRNAs with significantly altered translation efficiencies following inhibitor treatment (FDR < 0.05 and Log2 FC > 0.25). (**E**) Numbers of differentially translated mRNAs (ΔTE) in each quartile of genes (stratified by TE in DMSO) in cells treated with C6 (left) or C16 (right). (**F**) Enrichment analysis of common mRNAs suppressed by C6/

*Figure 2 continued on next page*

*Figure 2 continued*

C16 at the mRNA (blue) and translational (red; TE) level in MV4;11 cells. Hallmark.MSigDB pathways are shown. The x-axis indicates the number of suppressed genes in each category; the italic numbers are the corresponding FDR. See *Figure 2—source data 2* for the full Hallmark.MSigDB analysis, as well as for Reactome and KEGG pathways. (**G**) Enrichment analysis of mRNAs suppressed translationally by C6/C16 but with no significant changes in mRNA levels. Gene Ontology (GO) Biological Process (BP) and Molecular Function (MF) categories are shown, as well as KEGG pathways. The x-axis displays -Log10 FDR; the number of mRNAs is shown in italics in each bar. See *Figure 2—source data 3* for extended enrichment analyses, broken down by TE and mRNA direction changes. (**H**) TE changes in WDR5-bound (left) and non-bound (right) RPGs elicited by C6 (top) or C16 (bottom).

The online version of this article includes the following source data and figure supplement(s) for figure 2:

**Source data 1.** Output of Ribo-seq analysis of MV4;11 cells treated with C6/C16.

**Source data 2.** Hallmark, Reactome, and KEGG enrichment analysis of differentially translated genes in Ribo-seq.

**Source data 3.** Enrichment analysis of differentially translated genes, broken down by mRNA level change direction.

**Figure supplement 1.** WIN site inhibitors (WINi) suppress bulk protein synthesis.

**Figure supplement 2.** WIN site inhibitors (WINi) suppress translation.

**Figure supplement 3.** WIN site inhibitors (WINi) impair translation of mitochondrial ribosomal proteins.

in ribosomal protein levels at two timepoints: 24 hr, when there is no overt cellular response to WIN site inhibition, and 72 hr, when cell proliferation begins to be inhibited (*Aho et al., 2019a*). The abundance of ribosomal proteins allows for the use of label-free quantitative mass spectrometry (LFQMS; *Cox et al., 2014*) in whole-cell lysates to feasibly track ribosome protein levels, while also providing insight into other changes in protein levels promoted by WINi. Spike-in controls were not included.

In this analysis, we tracked ~3,200 proteins at each timepoint (*Figure 3A*, *Figure 3—source data 1*), ~850 of which are significantly altered by C16 treatment. Consistent with the subtle effects of WINi on mRNA abundance and TE, most differences in protein levels triggered by C16 are less than twofold in magnitude (*Figure 3B and C*). At 24 hr, ~90% of the proteins that change in response to C16 score as increased, whereas by 72 hr this number drops to ~60% (*Figure 3A*). Although the induction of proteins in response to C16 is unexpected for agents that decrease translational capacity, we note that protein synthesis is largely unaffected after 24 hr of C16 treatment (*Figure 2—figure supplement 1*), and that this phenomenon is unlikely to be an artifact of normalization. Indeed, the distribution of peptide intensities in the LFQMS data is unaffected by normalization (*Figure 3—figure supplement 1A*), and we see almost as many highly abundant proteins increasing as decreasing with C16 treatment (*Figure 3—figure supplement 1B–D*), arguing against the idea that large decreases in highly expressed proteins (such as the RPs; *Figure 3—figure supplement 1B*) are creating the appearance of less abundant proteins being induced. Instead, we suggest that this may be a transient compensatory mechanism, or an early part of the response to WINi. Regardless, most instances of increased protein levels are transient—fewer than one-third of these proteins are still induced at day 3 (*Figure 3D*)—whereas a majority of the proteins decreased at day 1 are also decreased at day 3. Enrichment analysis (*Figure 3—figure supplement 2A*, *Figure 3—source data 2*) reveals that proteins induced at 24 hr are modestly enriched in those connected to exocytosis and leukocyte activation, as well as mTORC1 signaling and MYC. By 72 hr, we see induction of proteins linked to glycolysis and fatty acid metabolism, as well as apoptosis. Additionally, manual curation reveals that the number of induced p53 target proteins increases over time: 13 are induced at 24 hr compared to 24 at the 3-day point (*Figure 3E*). Commensurate with the onset of a functional response to WINi, therefore, is a modest expansion in the apparent impact of p53 on the proteome, as well as the emergence of apoptotic response indicators.

Not surprisingly, proteins that are reduced in abundance at 24 hr are significantly enriched in those linked to the ribosome (*Figure 3—figure supplement 2B*). This enrichment becomes stronger at 72 hr. We also observe, at 72 hr, suppression of proteins linked to MYC and E2F targets, as well as mTORC1 signaling. In terms of ribosome components, this analysis reveals a progressive decline in the ribosomal protein inventory. Going from 24 to 72 hr, there is an increase in the number of impacted ribosomal subunits as well as in the magnitude of their suppression (*Figure 3F*), and eventually almost all ribosomal subunits are in deficit, regardless of whether or not they are encoded by a WDR5-bound gene (*Figure 3G*). RPL22L1 is the most strongly suppressed protein at 72 hr (*Figure 3C*), with its levels reduced by an order of magnitude after 3 d of C16 treatment. Consistent with the highly coordinated nature of ribosome biogenesis (*Dörner et al., 2023*), decreases in the abundance of RP

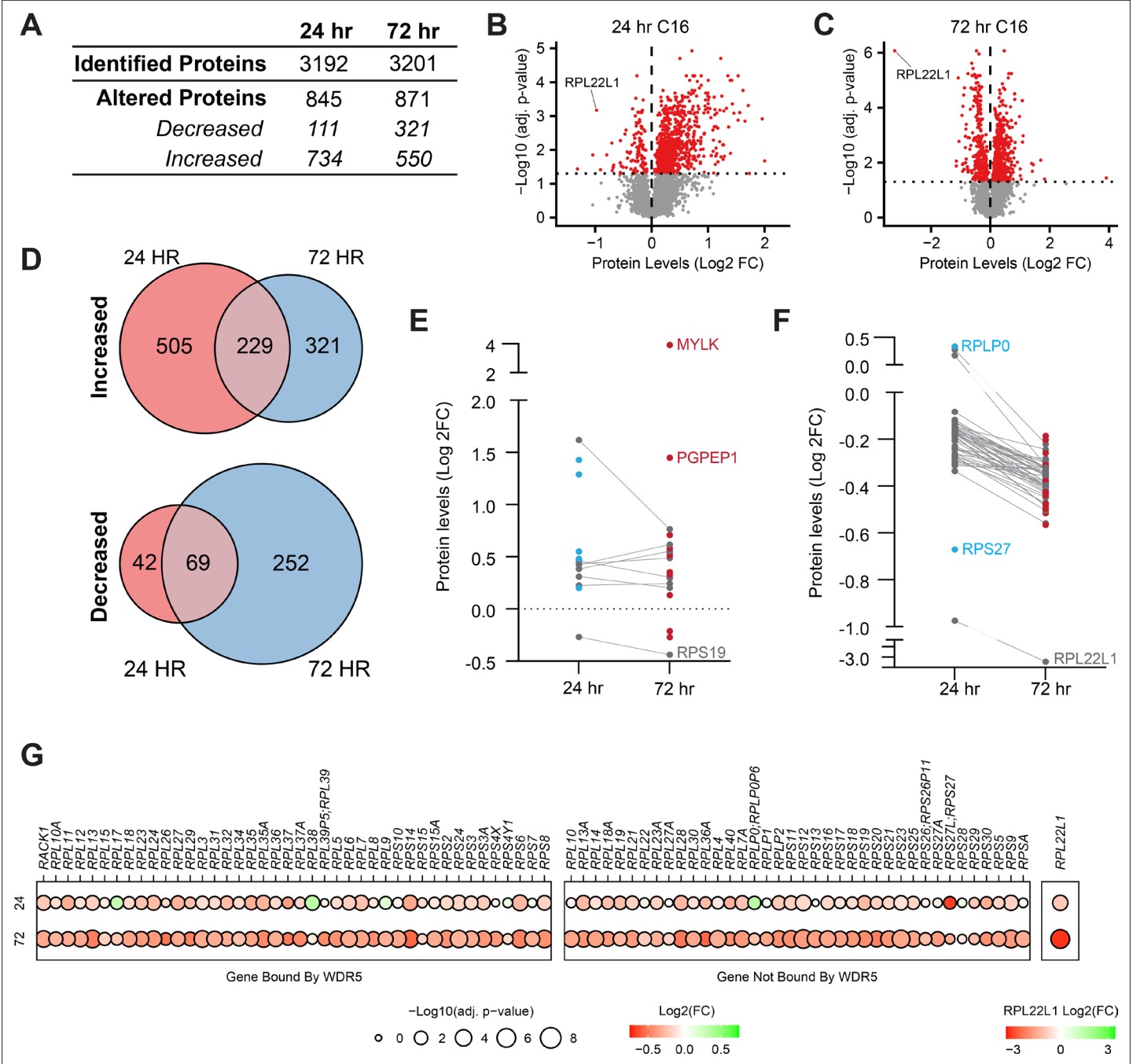

**Figure 3.** Impact of WIN site inhibitors (WINi) on the ribosome inventory of MLLr cancer cells. (**A**) Lysates from MV4;11 cells treated 24 or 72 hr with either 0.1% DMSO or 250 nM C16 were subjected to liquid chromatography coupled with tandem mass spectrometry and analyzed by label-free quantification (LFQMS). The table shows the number of proteins detected in DMSO and C16 samples and those with significantly altered levels at each time point (n = 4; adj. p-value<0.05). See ***Figure 3—source data 1*** for complete output of LFQMS analysis. (**B**) Volcano plot, showing protein level alterations in cells treated with C16 for 24 hr (red indicates adj. p-value<0.05). The location of RPL22L1 is indicated. (**C**) As in (**B**) but for 72 hr treatment with C16. (**D**) Overlap of proteins significantly increased (top) or decreased (bottom) following 24 or 72 hr C16 treatment. (**E**) Protein level alterations induced by C16 in consensus p53 target proteins (***Fischer, 2017***) at the 24 and 72 hr treatment timepoints. Those proteins only altered in abundance at 24 hr are represented as blue dots; proteins only altered at 72 hr are red; proteins altered at both timepoints are gray. (**F**) As in (**E**) but for ribosomal proteins. (**G**) Changes in expression of proteins encoded by WDR5-bound (left) and non-bound (right) RPGs elicited by 24 (top) or 72 (bottom) hr treatment with C16. Note that, due to the magnitude of change, Log2(FC) for RPL22L1 is presented on a separate scale.

The online version of this article includes the following source data and figure supplement(s) for figure 3:

**Source data 1.** Output of label-free quantitative mass spectrometry (LFQMS) analysis of MV4;11 cells treated with C16.

*Figure 3 continued on next page*

*Figure 3 continued*

**Source data 2.** Enrichment analysis of proteins altered in abundance by 24 or 72 hr of C16 treatment.

**Figure supplement 1.** Distribution of peptide/protein intensities in label-free quantitative mass spectrometry (LFQMS) analysis.

**Figure supplement 2.** Enrichment analysis of proteins with altered expression in response to C16 treatment.

**Figure supplement 3.** WIN site inhibitors suppress rRNA levels.

**Figure supplement 3—source data 1.** Raw unprocessed gel images corresponding to *Figure 3—figure supplement 3A*.

**Figure supplement 4.** C16 induces redistribution of nucleophosmin from the nucleolus to the nucleoplasm.

are accompanied by a progressive decline in rRNA expression—as revealed by metabolic labeling of RNAs with 2'-azido-2'-cytidine (AzCyd) (*Figure 3—figure supplement 3*, *Figure 3—figure supplement 3—source data 1*). Together, these experiments reveal that changes in ribosomal protein levels predicted from our transcriptomic studies manifest in reduced expression of ribosome components. Contrary to our earlier idea that WINi promote ribosome subunit imbalance, however, these data support a simpler model in which these inhibitors ultimately induce attrition of the majority of ribosomal proteins—as well as mature rRNAs.

Finally, we asked if the decline in ribosome inventory triggered by C16 is associated with nucleolar stress, as we have shown with C6 (*Aho et al., 2019a*). We used immunofluorescence to measure the redistribution of nucleophosmin (NPM1) from the nucleolus to the nucleoplasm; a characteristic of this phenomenon (*Russo and Russo, 2017*). Because inhibition of ribosome biogenesis via some rRNA inhibitors can induce a DNA damage response (*Sanij et al., 2020*), we also probed for the DNA damage marker γ-H2AX. We observed no obvious change in nucleolar morphology with up to 72 hr of C16 treatment (*Figure 3—figure supplement 4A*). We did, however, see a significant decrease in the nucleolar enrichment of NPM1 at the 72 hr treatment timepoint (*Figure 3—figure supplement 4B*), indicative of a nucleolar stress response. Notably, we did not observe induction of γ-H2AX foci in either the nucleolus or nucleoplasm, detecting it only in cells that were morphologically apoptotic (*Figure 3—figure supplement 4A*), consistent with studies showing that γ-H2AX is induced via the DNA fragmentation that occurs during apoptosis (*Rogakou et al., 2000*). Based on this analysis, we conclude that activation of nucleolar stress occurs in response to prolonged exposure to C16 and that, compared to some rRNA inhibitors, widespread induction of DNA damage is not a specific consequence of the action of WINi.

## A loss-of-function screen for modulators of the response to WINi

Next, we conducted a two-tier loss-of-function screen to identify genes that modulate the response of MLLr cells to WINi (*Figure 4A*). Our objective was to compare C6 and C16, and to identify high-confidence hits that are disconnected from cell viability. In tier 1, we carried out a screen using the GeCKOv.2 sgRNA library (*Joung et al., 2017*), which targets ~19,000 genes with six sgRNAs each, as well as ~1,200 miRNAs (four sgRNAs each). After transducing the library into MV4;11 cells expressing Cas9, we treated for 2 wk with 2 μM C6, during which time rapidly growing cells emerged within the transduced population (*Figure 4—figure supplement 1A*). We harvested genomic DNA, performed next-generation sequencing, and compared sgRNA representation before and after C6 treatment. We then inventoried genes with significant enrichment/depletion in corresponding sgRNAs in the treated population, removed pan-essential genes (*Tsherniak et al., 2017*), and created a custom library in which non-essential protein-coding 'hits' are targeted by four different sgRNAs (*Doench et al., 2016*). The smaller tier 2 library was then screened against C6 or C16, this time against a parallel DMSO-treated control population. This two-tiered approach allowed us to efficiently screen two different WINi and identify hits that are validated (for C6 at least) with up to 10 unique sgRNAs.

Although the first tier did not discriminate between genes that modulate fitness and those that modulate WINi response, several interesting observations emerged. Guide RNAs corresponding to ~70 genes were enriched and ~675 were depleted (*Figure 4B*), most of the latter of which are pan-essential (*Figure 4—source data 1*). Satisfyingly, *TP53* is the most highly enriched gene in the screen (*Figure 4B*, *Figure 4—figure supplement 1B*). *CDKN2A* also scored as highly enriched in the initial screen (*Figure 4B*)—specifically those sgRNAs targeting p14$^{ARF}$(*Figure 4—figure supplement 1C and D*), an inhibitor of the p53 ubiquitin ligase MDM2 (*Sherr, 2001*). Further support for the importance of p53 is evidenced by network analysis (*Chang and Xia, 2023*) of the 27 miRNAs flagged as

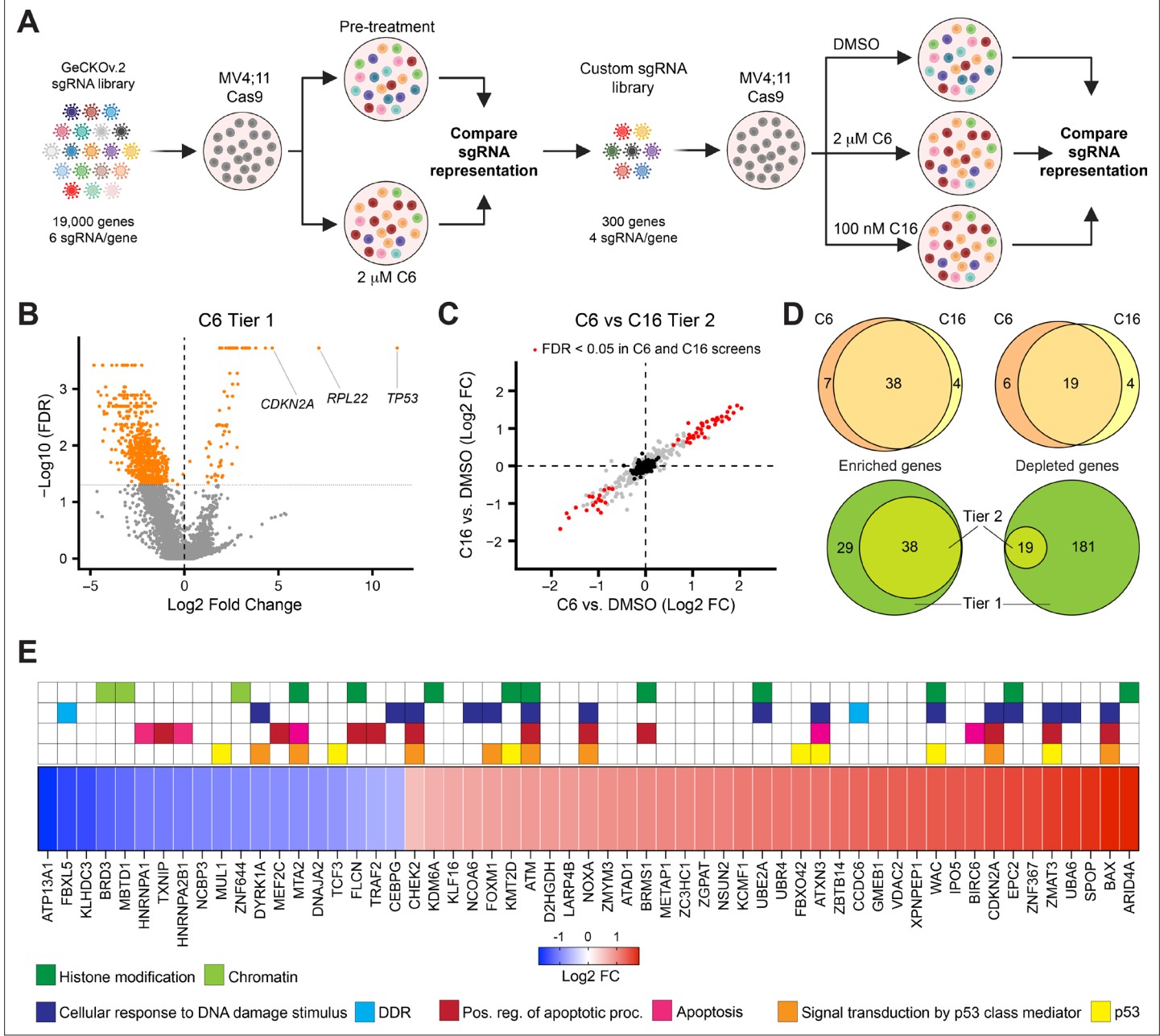

**Figure 4.** A two-tier loss-of-function screen for modulators of the response to WIN site inhibitors (WINi). (**A**) Two-tier screen design. In the first tier, Cas9-expressing MV4;11 cells were transduced with a genome-wide sgRNA library and treated with 2 μM C6 until a resistant cell population emerged. sgRNA representation in the pretreatment population was compared to the post-treatment population (n = 2). In the second tier, cells were transduced with a custom library of distinct sgRNAs targeting non-pan-essential 'hits' from the first tier, cultured in the presence of DMSO, C6, or C16, and sgRNA representation in C6/C16-treated cultures compared to that from DMSO-treated cultures (n = 2). (**B**) Volcano plot, showing gene-level changes in sgRNA representation from the first tier (orange indicates false discovery rate [FDR] < 0.05). Datapoints corresponding to *TP53*, *RPL22*, and *CDKN2A* are indicated. See ***Figure 4—source data 1*** for full output of the tier 1 screen. (**C**) Comparison of gene-level changes in sgRNA representation in C6- and C16-treated populations in the second tier screen, each compared to DMSO-treated populations (red indicates FDR < 0.05; black indicates non-targeting control sgRNAs). See ***Figure 4—source data 2*** for full output of the tier 2 screen. (**D**) Top: overlap of genes from the tier 2 screen with enriched (left) or depleted (right) sgRNAs in C6- and C16-treated MV4;11 populations, compared to the DMSO control. Bottom: overlap of genes with enriched (left) or depleted (right) sgRNAs in the first versus second tiers of the screen. 'Tier 1' contains only those genes targeted in the tier 2 screen. 'Tier 2' contains the intersection of genes with altered sgRNAs in both the C6 and C16 treatments. (**E**) Ranked heatmap, representing the mean gene-level Log2 fold change (FC) of sgRNAs from the C6 and C16 treatments in the tier 2 screen, as well as gene enrichment analysis outputs. Note that 'Signal transduction by p53 class mediator' is a GO:BP term (orange); 'p53' assignments (yellow) were added by manual curation.

*Figure 4 continued*

The online version of this article includes the following source data and figure supplement(s) for figure 4:

**Source data 1.** Output of the tier 1 screen.

**Source data 2.** Output of the tier 2 screen.

**Figure supplement 1.** Genome-wide CRISPR screen identifies genes that influence response to C6/C16.

enriched (*Figure 4—source data 1*), which display connections to p53 (*Figure 4—figure supplement 1E*). Finally, we note that the second most highly enriched gene in the first tier encodes a ribosomal protein: RPL22 (*Figure 4B*, *Figure 4—figure supplement 1F*). Because of the strong enrichment of sgRNAs against *RPL22* and *TP53*, we removed both genes from the second tier screen.

The second tier screen (*Figure 4—figure supplement 1G*, *Figure 4—source data 2*) revealed that the response of MV4;11 cells to C6 and C16 is very similar, both in terms of the enriched/depleted genes and their rankings (*Figure 4C*). A majority of genes that modulate the response to C6 similarly modulate the response to C16 (*Figure 4D*). For most of the depleted genes that appear specific to one WINi, similar depletion is observed with the other WINi, but is generally just over the FDR cutoff (*Figure 4—figure supplement 1H*). But for C6-specific enriched genes we see that most have high FDR values in the C16 samples, arguing that the earlier generation compound has expanded, off-target, activities. Gene Ontology (GO) enrichment analysis of the 57 common genes emerging from the screen revealed enrichment in four overlapping categories connected to p53 signaling, apoptosis, the DNA damage response (DDR), and histone modifications (*Figure 4E*, *Figure 4—figure supplement 1I*). The representation of genes connected to p53 and apoptosis reinforces the importance of p53-mediated cell death to the response of MLLr cells to WINi. We observe, for example, that loss of function of the p53 antagonist and ubiquitin ligase MUL1 (*Jung et al., 2011*), increases sensitivity to C6/C16, whereas loss of canonical p53 effectors NOXA, BAX, and ZMAT3 is associated with a decrease in response. The DNA damage response category overlaps with that of p53 but is nonetheless distinct and includes genes encoding the ATM and CHK2 kinases (*Blackford and Jackson, 2017*) and the FOXM1 transcription factor that activates DDR gene expression networks (*Zona et al., 2014*). This category also includes two depleted genes, encoding FBXL5—which antagonizes ATM signaling (*Chen et al., 2014*)—and DYRK1A—a kinase involved in the DDR (*Laham et al., 2021*), DREAM complex activation (*Litovchick et al., 2011*), and RPG transcription (*Di Vona et al., 2015*). The involvement of this category of enriched genes is intriguing, given the lack of γ-H2AX accumulation in non-apoptotic cells (*Figure 3—figure supplement 4*), and warrants further investigation in the future.

A majority of genes in the histone modification category, when disrupted, blunt the response to both WINi (*Figure 4E*). These genes include those encoding the H3K27 demethylase KDM6A (*Lan et al., 2007*), the MLL/SET protein KMT2D (MLL2; *Shinsky et al., 2015*), and ARID4A—a component of the mSin3/HDAC1 co-repressor complex (*Lai et al., 2001*). The most conspicuous sensitizing gene in this group is BRD3, a member of the BET family of proteins that includes BRD2 and BRD4 (*Eischer et al., 2023*). Interestingly, although BRD4 was not included in the second screen tier as it is pan essential, BRD2 was not included because it was not significantly enriched/depleted in the first tier (*Figure 4—source data 1*), revealing that the actions of BRD3 in modulating response to WINi are not shared with all family members. Further supporting the importance of BRD3 to the response, we note that SPOP, which targets BET family proteins for proteasomal destruction (*Janouskova et al., 2017*), is one of the most significantly enriched hits from the screen (*Figure 4E*).

Collectively, these findings demonstrate functional involvement of the ribosomal protein RPL22 in the response to WINi and confirm the importance of the p53 network to robust inhibition of MLLr cell growth by these agents. These findings also identify a number of candidate predictive biomarkers of response that can be further interrogated for their value as patient selection criteria.

## Identification of agents that synergize with WINi in MLLr cells

Given the ways in which resistance to WINi can arise (*Figure 4*), the most efficacious application of these agents will likely be in combination with other therapies. We therefore asked whether C16

synergizes with 11 approved or targeted agents. Several of the agents were chosen based on the results of our screen. Harmine—an inhibitor of the DYRK1A kinase (*Göckler et al., 2009*)—and the BET family inhibitor mivebresib (*Lin et al., 2017*) each target the product of sensitizing genes, whereas venetoclax inhibits BCL-2 (*Souers et al., 2013*)—an inhibitor of BAX, which scored as a resistance gene. We also tested agents connected to the DDR (etoposide, olaparib, and the ATR inhibitor VE821; *Charrier et al., 2011*), protein synthesis and homeostasis (alvespimycin and rapamycin), and p53 (nutlin-3a). Due to the enrichment of PRMT5 substrates in our translational profiling (*Figure 2— figure supplement 2F*), we queried the PRMT5 inhibitor pemrametostat (*Chan Penebre et al., 2015*). Because DOT1L inhibitors suppress not only classic MLL fusion target genes (*Bernt and Armstrong, 2011*; *Daigle et al., 2011*) but also RPGs (*Lenard et al., 2020*), we tested for synergy with the DOT1L inhibitor pinometostat (*Daigle et al., 2011*). We treated MV4;11 cells with a dose matrix spanning 49 unique dose combinations and quantified synergy δ-scores using the zero interaction potency (ZIP) model (*Yadav et al., 2015*; *Figure 5A and B*, *Figure 5—figure supplement 1A and B*, and *Figure 5—source data 1*).

In MV4;11 cells, we observe synergy with mivebresib, pemrametostat, pinometostat, etoposide, harmine, and venetoclax. Within this group are three agents selected based on sensitizing targets from the CRISPR screen, providing additional support for the role of DRYK1A, BCL-2/BAX, and BRD3 in the responsiveness to WINi. Of the three agents connected to the DDR, only etoposide displays significant synergy. Agents that target protein synthesis and homeostasis yield mixed results—we observe potent antagonism with the mTOR inhibitor (*Raught et al., 2001*) rapamycin (peak δ-score –16), while the HSP90 inhibitor alvespimycin (*Schnur et al., 1995*) is either antagonistic or synergistic, depending on dose (*Figure 5A*). Finally, we note that—of the agents displaying synergy—four are particularly strong (peak δ-scores > 10) and observed at agent doses consistent with on-target activity (*Figure 5—figure supplement 1B*), suggesting that mivebresib, pemrametostat, pinometostat, and venetoclax should be prioritized for in vivo testing. Focusing on these agents is further justified by our finding that all four are synergistic with C16 in MOLM13 cells (*Figure 5C and D*, *Figure 5—figure supplement 2A and B*).

To understand how combination with another agent impacts the response to WINi, we transcriptionally profiled MV4;11 cells treated for 48 hr with C16 and mivebresib, either as single agents or in combination, at concentrations that yield peak synergy between them (100 nM C16 and 2.5 nM mivebresib). Spike-in controls were not included. By RNA-seq (*Figure 5—source data 2*), it is clear that the functional synergy between C16 and mivebresib is apparent at the transcript level, with more than 6,200 gene expression changes in the combination treatment, compared to less than 1,800 for C16 and 2700 for mivebresib (*Figure 5E*). Notable are the very distinct transcriptional profiles induced by each agent alone, with fewer than 200 shared gene expression changes in each direction (*Figure 5F*). The impact on RPG expression of both agents is additive (*Figure 5—figure supplement 3A*), but in general we find that the combination of C16 and mivebresib dysregulates similar categories of genes for each agent alone, but with substantially more genes in each category (*Figure 5—source data 3*). This is clear for genes linked to translation (*Figure 5G*), p53 (*Figure 5—figure supplement 3B*), and the induction of apoptosis (*Figure 5H*). Thus, although further investigation is needed, this analysis is consistent with the idea that synergy between C16 and mivebresib results from alterations in the expression of distinct but complementary sets of genes that ultimately conspire to augment induction of p53.

## WINi inactivate MDM4 in an RPL22-dependent manner

Despite the importance of p53 in the response to WINi, WINi cause only a slight increase, if any, in p53 levels (*Figure 6—figure supplement 1A*, *Figure 6—source data 1*; *Aho et al., 2019a*). Interestingly, inactivation or loss of *RPL22* in cancer is associated with increased expression of *RPL22L1* and inclusion of exon 6 in MDM4 (*Ghandi et al., 2019*), an event that promotes MDM4 expression by preventing formation of a 'short' MDM4 mRNA isoform (MDM4s) that is destroyed by nonsense-mediated decay (*Rallapalli et al., 1999*). MDM4 is intriguing because it can suppress p53 without altering its stability (*Francoz et al., 2006*). It is also intriguing because skipping of exon 6 in the *MDM4* mRNA is stimulated by ZMAT3 (*Bieging-Rolett et al., 2020*; *Muys et al., 2021*) and antagonized by RPL22L1 (*Larionova et al., 2022*)—two genes that are oppositely regulated by WINi. We therefore asked if WINi induce changes in the levels of mRNA splice isoforms and if this includes MDM4.

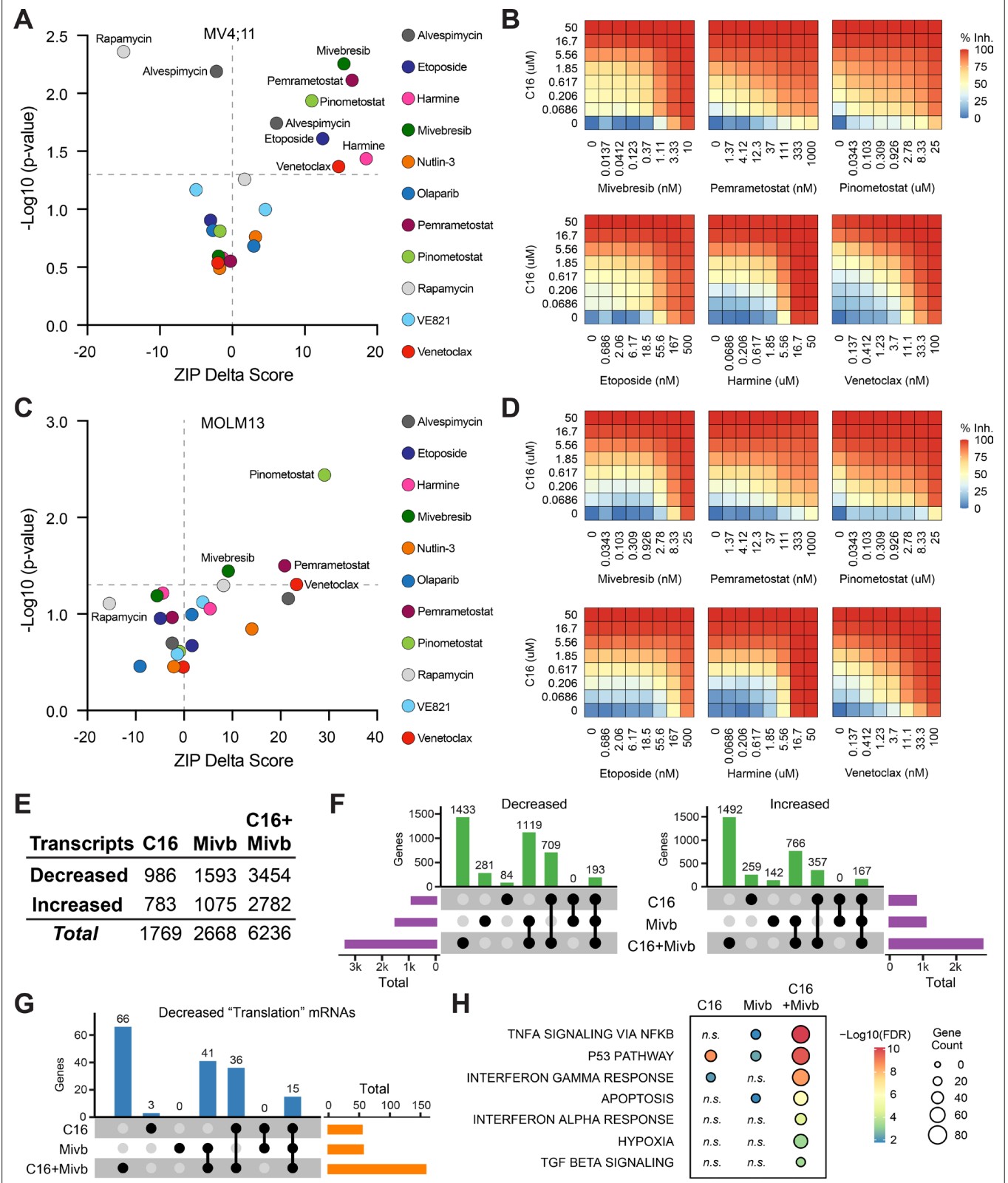

**Figure 5.** Identification of agents that synergize with WIN site inhibitors (WINi) in MLLr cells. (**A**) Peak synergy (>0) and antagonism (<0) zero interaction potency (ZIP) delta (δ) scores from synergy assays in which MV4;11 cells were treated for 3 d with 49 unique dose combinations of C16 and the indicated compound of interest (n = 4). See *Figure 5—source data 1* for numerical ZIP delta analysis output. (**B**) Heatmaps of MV4;11 cell growth inhibition at each dose of C16 and the indicated six compounds. The remaining five combinations tested are shown in *Figure 5—figure supplement 1*. (**C**) As in

*Figure 5 continued on next page*

*Figure 5 continued*

(**A**) but for MOLM13 cells. See *Figure 5—source data 1* for numerical ZIP delta analysis output. (**D**) As in (**B**) but for MOLM13 cells. The remaining five combinations tested are shown in *Figure 5—figure supplement 2*. (**E**) Number of genes with significantly (false discovery rate [FDR] < 0.05) altered transcript levels following treatment of MV4;11 cells with C16 (100 nM), mivebresib (Mibv; 2.5 nM), or the combination for 48 hr, as determined by RNA-seq (n = 3). See *Figure 5—source data 2* for complete output of RNA-seq analysis. (**F**) UpSet plot, showing the overlap of genes suppressed (left) or induced (right) in response to C16, mivebresib, or the combination. (**G**) UpSet plot, showing the breakdown of Reactome 'Translation' pathway genes suppressed in response to C16, mivebresib, or the combination. (**H**) Enrichment of Reactome Pathways in genes with increased transcripts following treatment of MV4;11 cells with C16, mivebresib, or the combination. See *Figure 5—source data 3* for complete output of enrichment analyses.

The online version of this article includes the following source data and figure supplement(s) for figure 5:

**Source data 1.** Peak synergy and antagonism scores for MV4:11 and MOLM13 cells treated with C16 in combination with 11 agents.

**Source data 2.** Output of RNA-seq analysis of MV4;11 cells treated with C16, mivebresib, or both.

**Source data 3.** Enrichment analysis of differentially expressed genes in RNA-seq of MV4;11 cells treated with C16, mivebresib, or both.

**Figure supplement 1.** C16 is synergistic with multiple agents in MV4;11 cells.

**Figure supplement 2.** C16 is synergistic with multiple agents in MOLM13 cells.

**Figure supplement 3.** Impact of C16 and mivebresib on RPG and p53 target gene expression.

RNA-seq data (*Figure 1*) were interrogated for alternative splicing events (*Shen et al., 2014*). At an FDR < 0.05 and a threshold of ≥5% change in exon inclusion ($\Delta\psi$), C6 and C16 each result in changes in ~1,000 differentially spliced mRNAs (*Figure 6—figure supplement 1B*), ~250 of which are shared between the two inhibitors (*Figure 6A*). Many of these changes reflect events with low read counts or at minor splice sites (*Figure 6—source data 2*), representative examples of which are presented in *Figure 6—figure supplement 1C and D*. That said, WINi clearly promote accumulation of *MDM4* transcripts in which exon 6 is skipped (*Figure 6B*). We also observe splicing changes at *RPL22L1* itself (*Figure 6—figure supplement 1E*), where WINi leads to the depletion of transcripts in which exon 2 is spliced to a distal 3′ acceptor site in exon 3. This splicing event encodes the RPL22L1a isoform that is incorporated into ribosomes (*Larionova et al., 2022*). Splicing to the proximal 3′ acceptor site, which generates a non-ribosomal RPL22L1b isoform that modulates splicing, is insensitive to WINi. We confirmed the impact of C16 on *MDM4* and *RPL22L1* splice isoforms by semi-quantitative RT-PCR (*Figure 6—figure supplement 1F*, *Figure 6—source data 3*) and quantitative RT-PCR (*Figure 6—figure supplement 1G*). Based on these observations, we conclude that treatment of MLLr cells with WINi promotes the selective loss of transcripts encoding RPL22L1a and MDM4.

The association of *RPL22* loss with increased expression of *RPL22L1* and inclusion of exon 6 in *MDM4* (*Ghandi et al., 2019*) prompted us to ask how *RPL22* contributes to the response of MLLr cells to WINi. Knockout (KO) of *RPL22* (*Figure 6—figure supplement 2A*, *Figure 6—source data 4*) decreases the sensitivity of MV4;11 and MOLM13 cells to C16 by three- to fivefold compared to non-targeted (NT) control cells (*Figure 6C*, *Figure 6—figure supplement 2B*), as well as attenuating the modest induction of p53 protein observed in the MV4;11 line (*Figure 6D*, *Figure 6—source data 5*). The response of relatively insensitive (p53-null) K562 cells, in contrast, is unaffected by *RPL22* disruption (*Figure 6C*). RNA-seq analysis, performed without spike-in controls, (*Figure 6—figure supplement 2C and D*, *Figure 6—source data 6*) reveals that disruption of *RPL22* does not impact the effect of WINi on WDR5-bound RPGs (*Figure 6—figure supplement 2E*), but it does block the effects of C16 on expression of *RPL22L1* and *ZMAT3* (*Figure 6—figure supplement 2D*), as well as tempering its ability to suppress genes connected to the cell cycle, mTORC1 signaling, and MYC (*Figure 6—figure supplement 2F*, *Figure 6—source data 7*). Notably, *RPL22* loss also impairs induction of genes involved in p53 signaling (*Figure 6E*, *Figure 6—figure supplement 2G*). We also observe that mitochondrial RPGs are induced by WINi uniquely in *RPL22*-null cells (*Figure 6—figure supplement 2H*). We conclude that RPL22 is needed for a majority of the characteristic responses of MLLr cells to WINi, including activation of p53.

Finally, we asked if *RPL22* knockout alters patterns of alternative splicing induced by WINi. Thousands of differences were detected in splice isoforms between the various pairwise comparisons (*Figure 6—source data 8*). In general, *RPL22KO* cells show fewer C16-induced changes in alternative splicing patterns than NT cells (*Figure 6—figure supplement 3A*). As we observed above, a majority of the changes reflect events with low read counts or at minor splice sites, with two notable exceptions: *MDM4* and *RPL22L1*. In the absence of WINi, disruption of *RPL22* promotes exon 6 retention

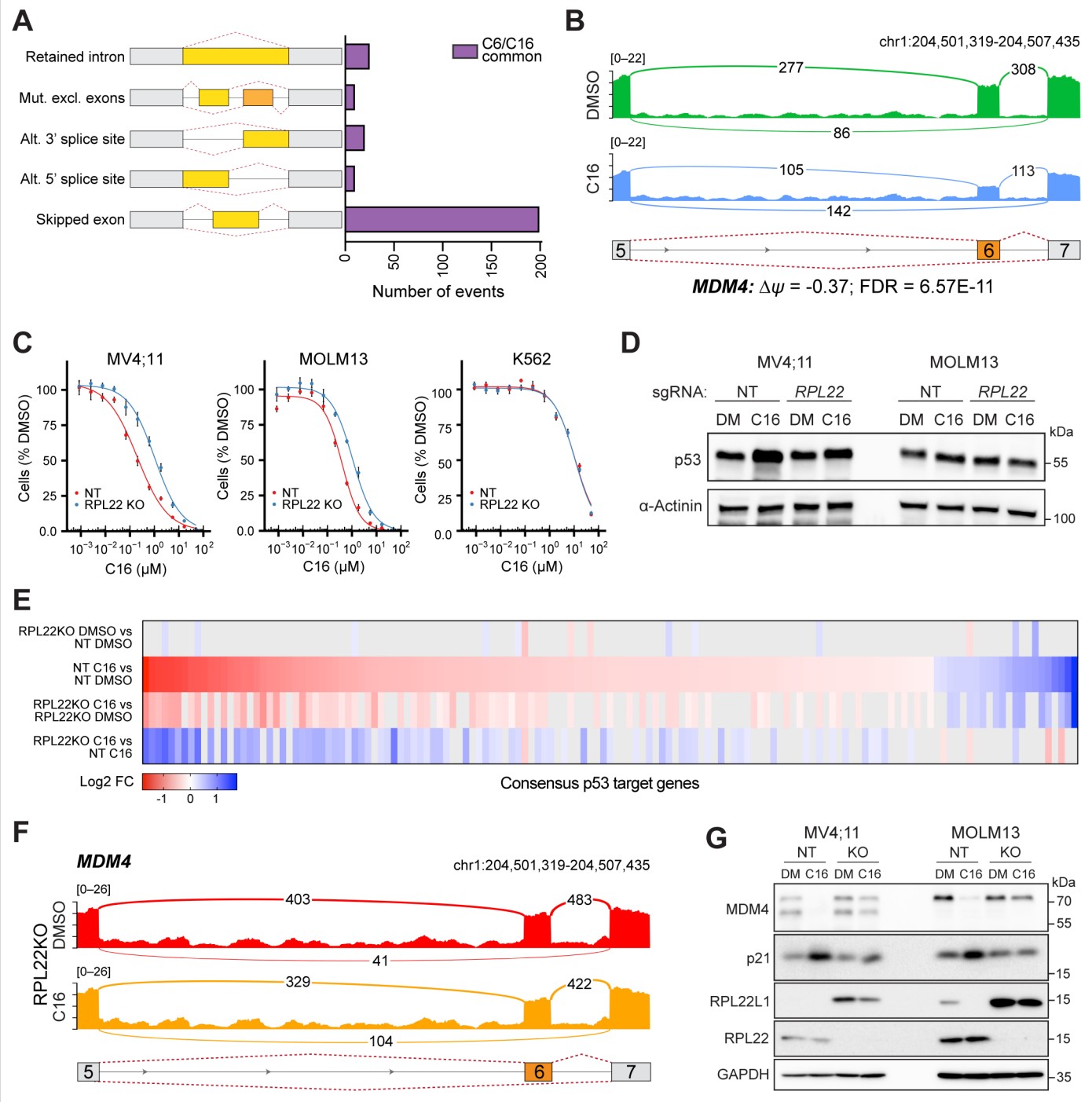

**Figure 6.** WIN site inhibitors (WINi) inactivate MDM4 in an RPL22-dependent manner. (**A**) Differential alternative splicing events affected by C6/C16 treatment of MV4;11 cells were quantified by rMATS. The types of alternative splicing events are cartooned at left, and the number of significantly different events (>5% Δψ; false discovery rate [FDR] < 0.05) common to C6/C16 depicted in the graph. See *Figure 6—source data 2* for output of rMATS analysis. (**B**) Sashimi plot quantifying read junctions that span exons 5–7 of *MDM4* in MV4;11 cells treated with DMSO (green) or C16 (blue). Numbers in the arcs display junction depth. The location of exons 5, 6, and 7 is depicted at the bottom; skipped exon 6 is highlighted in orange. (**C**) Viabilities of control (non-targeting: NT) and RPL22 knock out (KO) MV4;11, MOLM13, and K562 cells treated with a serial dilution range of C16 for 72 hr, relative to viability of DMSO-treated cells (n = 3; mean ± SEM). (**D**) Western blot analysis of p53 levels in control (NT) and RPL22 knockout (KO) MV4;11 and MOLM13 cells treated with either 0.1% DMSO or C16 (MV4;11, 200 nM; MOLM13, 400 nM) for 72 hr. α-Actinin is loading control. Representative images from three biological replicates shown. Raw unprocessed gel images are presented in *Figure 6—source data 5*. (**E**) Heatmap, showing significant changes in the expression of consensus p53 target genes (*Fischer, 2017*) between the indicated pairwise comparisons of RNA-seq datasets. Note that only consensus p53 target genes altered in expression by C16 in control (NT) cells are represented. (**F**) Sashimi plot quantifying read

*Figure 6 continued on next page*

*Figure 6 continued*

junctions that span exons 5–7 of *MDM4* in RPL22KO MV4;11 cells treated with DMSO or C16. Numbers in the arcs display junction depth. The location of exons 5, 6, and 7 is depicted at the bottom; skipped exon 6 is highlighted in orange. Corresponding NT images are presented alongside RPL22KO images in *Figure 6—figure supplement 3B*. (G) Western blots, comparing the effects of 72 hr of DMSO (DM) or C16 treatment (MV4;11, 200 nM; MOLM13, 400 nM) of control (NT) or RPL22 knockout (KO) MV4;11 (left) or MOLM13 (right) cells on levels of MDM4, p21, RPL22L1, RPL22, and GAPDH (loading control). Representative images from three biological replicates are shown. Raw unprocessed gel images are presented in *Figure 6—source data 9*.

The online version of this article includes the following source data and figure supplement(s) for figure 6:

**Source data 1.** Raw unprocessed gel images corresponding to *Figure 6—figure supplement 1A*.

**Source data 2.** Output of rMATS analysis of MV4;11 cells treated with C6/C16.

**Source data 3.** Raw unprocessed gel images corresponding to *Figure 6—figure supplement 1F*.

**Source data 4.** Raw unprocessed gel images corresponding to *Figure 6—figure supplement 2A*.

**Source data 5.** Raw unprocessed gel images corresponding to *Figure 6D*.

**Source data 6.** Output of RNA-seq analysis of NT and RPL22KO MV4;11 cells treated with C16.

**Source data 7.** GSEA Hallmark and GOBP enrichment analysis of differentially expressed genes in RNA-seq of NT and RPL22KO MV4;11 cells treated with C16.

**Source data 8.** Output of rMATS analysis of NT and RPL22KO MV4;11 cells treated with C16.

**Source data 9.** Raw unprocessed gel images corresponding to *Figure 6G*.

**Figure supplement 1.** WIN site inhibitors (WINi) alter the abundance of alternatively spliced mRNA isoforms.

**Figure supplement 2.** Impact of RPL22 loss on the response of MLLr cells to WIN site inhibitors (WINi).

**Figure supplement 3.** Impact of RPL22 loss on the abundance of alternatively-spliced mRNA isoforms in MV4;11 cells.

in *MDM4* (*Figure 6—figure supplement 3B*) and induces expression of the mRNA splice isoform encoding RPL22La (*Figure 6—figure supplement 3C*). In the presence of C16, disruption of *RPL22* mitigates skipping of exon 6 in *MDM4* (*Figure 6F*) and largely blocks suppression of the RPL22L1a-encoding mRNA isoform (*Figure 6—figure supplement 3C*). Importantly, these changes in mRNA isoforms manifest at the protein level (*Figure 6G*, *Figure 6—source data 9*) as we observe that both MDM4 and RPL22L1 protein expression is suppressed by C16, that RPL22L1 is induced by RPL22 disruption, and that loss of RPL22 blocks a majority of the ability of C16 to reduce expression of the MDM4 and RPL22L1 proteins. We also note that loss of RPL22 blocks induction of the p53 target, p21. Taken together, these data demonstrate that RPL22 loss can cause the induction of RPL22L1 and inclusion of exon 6 in MDM4 observed in cancer (*Ghandi et al., 2019*) and reveal that splicing-driven suppression of MDM4 is an important mechanism through which WINi activate p53 in MLLr cells.

## Discussion

Here, we describe an integrated multi-omic approach to characterize the mechanism of action of WDR5 WINi in MLL-rearranged cancer cells. By combining transcriptional, translational, and proteomic profiling with genome-wide loss-of-function screens, we demonstrate the broad impact of WINi on the ribosomal protein complement and translational capacity of MLLr cells, reveal the importance of multiple arms of the p53 response pathway in cellular inhibition, and uncover a role for alternative splicing of MDM4 in activating p53 in this setting. Collectively, these data cast WINi as a novel ribosome-directed anticancer therapy and provide insight into patient selection criteria, mechanisms of resistance, and strategies to improve WINi efficacy in the clinic.

The primary targets of WINi, we propose, are the cohort of ~40 RPGs that are bound by WDR5 in all cell types examined (*Aho et al., 2019a*; *Bryan et al., 2020*; *Florian et al., 2022*). These RPGs are rapidly transcriptionally suppressed in response to WIN site blockade (*Bryan et al., 2020*; *Florian et al., 2022*; *Aho et al., 2019a*) or WDR5 degradation (*Siladi et al., 2022*), and show sustained suppression at the mRNA level. These same RPGs are suppressed by disruption of the MYC–WDR5 interaction (*Thomas et al., 2019*), implying that the function of WDR5 at these genes is to recruit MYC, a prominent target of MLL-fusion oncoproteins (*Ahmadi et al., 2021*). Previously, we posited that suppression of half the RPGs would lead to a ribosomal subunit imbalance (*Aho et al., 2019a*), but our analysis shows that WINi deplete the entire ribosomal inventory. Part of this depletion is driven

by decreased translation of ribosomal mRNAs, although it is also possible that ubiquitin-mediated quality control mechanisms or ribophagy (*Zhao et al., 2022*) degrade ribosomal proteins under these conditions.

The effects of WINi on ribosomal protein levels are extensive in terms of the number of impacted subunits, but not the magnitude of change, which caps at around a 30% decrease by 72 hr. This cap could be set by the maximal contribution of MYC–WDR5 to the expression of target RPGs, which is twofold as determined by genetic disruption of the MYC–WDR5 interaction (*Thomas et al., 2019*), degradation of WDR5 (*Siladi et al., 2022*), or WIN site blockade. Although modest, it should be noted that a 30% decrease in ribosomal protein levels corresponds to a loss of up to 3 million ribosomes per cell (*Shore and Albert, 2022*), and in that light it is not surprising that WINi result in reduced translational efficiencies of about half of all actively translated mRNAs. Unlike perturbations such as ER stress (*Advani and Ivanov, 2019*), WINi does not promote preferential translation of stress-responsive mRNAs, although there is some specificity in terms of the translational consequences. Messenger RNAs carrying 5'TOP motifs, for example, are spared from the full translational impact of C6/C16, and we see distinct biological clustering of mRNAs with decreased translation efficiency. Whether these patterns are intrinsic to WINi, or a general response of MV4;11 cells to translational stress, remains to be determined. Overall, we conclude that WINi do not trigger extensive translational reprogramming, but rather act to induce a widespread yet restrained translational choke.

In addition to ribosomal protein attrition, we also find that WINi trigger a progressive decline in rRNA expression, which we suggest is a secondary effect resulting from the tight coordination between RPG and rRNA transcription (*Dörner et al., 2023*). As previously reported (*Aho et al., 2019a*), we confirm that they promote a shift in the relative abundance of NPM1 in the nucleoplasm versus the nucleolus, indicative of a nucleolar stress response. Given that we recovered multiple DDR components in our two-tier CRISPR screen, and that agents such as the RNA polymerase I inhibitor CX-5461 induce DNA damage (*Quin et al., 2016*; *El Hassouni et al., 2019*), we had expected to see some evidence of γ-H2A.X induction in the nucleolus or the nucleoplasm in response to WINi. Yet we only observe γ-H2A.X induction in apoptotic cells. The difference in this aspect of the response to what are overtly two different ways to inhibit protein synthesis in cancer cells—RPG versus rRNA inhibition—may relate to the different primary mechanism of action of these agents or the magnitude of their effects, which are more subtle with WINi. Alternatively, because not all rRNA inhibitors induce DNA damage (e.g., *Peltonen et al., 2014*), induction of DNA damage may be specific to CX-5461. Further investigation, especially into the significance of DDR components in the response to WINi, is warranted.

Activation of p53 is, however, a major point of convergence of WDR5 and rRNA inhibitors. Not only do we detect activation of p53 target genes in response to WIN site blockade, but we observe synergistic activation of p53 target genes with the BET bromodomain inhibitor mivebresib and suppression of p53 targets upon deletion of *RPL22*; both of which correlate with cellular sensitivity. We also recover multiple components of the p53 signaling pathway as 'resistance' genes in our CRISPR screens, including p53 itself and the splicing factor ZMAT3. Inhibition of rRNA synthesis is thought to activate p53 by generating excess ribosomal proteins that bind to and inactivate MDM2 (*Pfister, 2019*). It is likely that inhibition of MDM2 contributes to p53 activation in response to C6/C16, as we see a modest increase in p53 levels in MV4;11 cells exposed to WINi, and find that loss of the MDM2 inhibitor p14$^{ARF}$ renders MV4;11 cells less sensitive to C6/C16. Here, however, we also find that there is a second route of p53 activation at work, in which WINi promote accumulation of alternatively spliced isoforms of *MDM4* mRNA in which exon 6 is skipped, driving down MDM4 protein levels. Because MDM4 inhibits p53 via proteolysis-independent mechanisms (*Francoz et al., 2006*), these findings explain how WINi can induce a robust p53 target gene signature in the absence of frank induction of p53 protein. They also point to a dominant role of the MDM4–p53 axis in mediating the response of MLLr cells to WIN site blockade. Whether rRNA inhibition triggers p53 activation by a similar mechanism has, to our knowledge, yet to be reported.

In considering the mechanism through which WINi inactivate MDM4, it is possible that the balance of RPL22L1 and ZMAT3, which are oppositely regulated in response to WIN site inhibition, governs the extent of MDM4 exon 6 inclusion. ZMAT3 is induced in response to p53 activation and promotes skipping of exon 6 in MDM4 (*Bieging-Rolett et al., 2020*). RPL22L1, in contrast, which is potently suppressed by WINi, promotes exon 6 inclusion (*Larionova et al., 2022*). Any process that tips the

balance in favor of ZMAT3, therefore, would be expected to inhibit MDM4, activate p53, and initiate a feed-forward mechanism that drives ZMAT3 expression, fortifies p53 induction, and commits cells to an apoptotic outcome. Induction of ZMAT3 alone should be sufficient to trigger this circuit, but the conspicuous suppression of RPL22L1 by WINi suggests that this may also contribute to the response. Paradoxically, the isoform of RPL22L1 that is suppressed by WIN site blockade, RPL22L1a, is linked in glioblastoma cells to ribosome function, not splicing (*Larionova et al., 2022*), while the splicing-relevant RPL22L1b isoform is resistant to WINi. One possibility is that, despite its recurrence and prominence, suppression of RPL22L1a by C6/C16 does not contribute to MDM4 suppression, and induction of ZMAT3 is the critical driving factor. Alternatively, RPL22L1a may indeed act to control splicing in MLLr cells, in contrast to what has been reported in glioblastoma. Further investigation is required.

The RPL22–RPL22L1–MDM4 nexus we encountered has been inferred by genome-wide studies of the Cancer Cell Line Encyclopedia (CCLE; *Ghandi et al., 2019*), and our work here demonstrates that loss of *RPL22* causes induction of RPL22L1 and promotes inclusion of exon 6 in MDM4. We have no evidence that RPL22 itself plays a role in the response to WINi in otherwise unperturbed MLLr cells. Rather, we suggest that its recovery as a resistance gene in our CRISPR screen is tied to its ability to suppress *RPL22L1* expression (*O'Leary et al., 2013*), and the ectopic effect of its deletion on preventing WINi-induced RPL22L1a decline. That said, RPL22 status is likely to be highly relevant in terms of patient selection criteria. *RPL22* is frequently inactivated or deleted in primary cancer samples (*Kandoth et al., 2013*; *Goudarzi and Lindström, 2016*; *Ghandi et al., 2019*), as well as 7% of lines in the CCLE (*Cao et al., 2017*). Unlike other RPGs, mutation or deletion of *RPL22* is not associated with mutational inactivation of p53, and indeed there is a strong tendency for wild-type p53 to be retained in *RPL22* mutant/deletion lines (*Cao et al., 2017*; *Ghandi et al., 2019*). In practical terms, therefore, cancers that retain wild-type p53 but otherwise are mutated/deleted for *RPL22*, or overexpress *RPL22L1*, would not be expected to robustly respond to WINi.

Although p53 is important for the action of WINi in MLLr cells, there are likely other stress response mechanisms that mediate cellular inhibition by these agents. A number of p53-independent nucleolar stress responses have been identified, but these remain mechanistically opaque compared to p53-dependent responses (*Boglev et al., 2013*; *Pfister et al., 2015*; *Jayaraman et al., 2017*). Conversely, we might also expect cells to be able to mount protective responses to WIN site blockade. In this regard, it is curious that four resistance genes identified in our CRISPR screen—UBA6, BIRC6, KCMF1, and UBR4—encode members of a newly identified BIRC6 ubiquitin-ligase complex (*Cervia et al., 2023*), the function of which is to prevent aberrant activation of the integrated stress response (ISR). The ISR is a central regulator of protein homeostasis (*Costa-Mattioli and Walter, 2020*) that drives protective translational reprogramming in response to multiple cellular stresses. There is no indication that the ISR is activated by WINi; indeed, the master regulator of ISR, ATF4, is suppressed by C6/C16 at the mRNA and translational levels. But the finding that loss of all four members of the BIRC6 complex blunts the response to WINi implies that ISR activation can be a mechanism through which cells evade the full impact of these agents.

As with most monotherapies, future single-agent WINi treatment paradigms are likely to encounter resistance either by activation of protective responses such as those proposed above or by mutations in one or more of the resistance genes recovered in our CRISPR screen. Identification of agents that can be used in combination with WINi to increase cancer cell inhibition is thus crucial. Our relatively limited synergy screening identified a number of combinations that should be prioritized for in vivo testing. We found that WINi act synergistically with the BCL-2 inhibitor venetoclax (*Souers et al., 2013*). This is rationalized by our recovery of BAX as a resistance gene and is noteworthy because venetoclax is an approved therapy for several blood-borne cancers. We also identified notable synergies with experimental agents targeting BET bromodomain family members, DOT1L, and PRMT5. The combination with mivebresib is rationalized based on identification of BRD3 as a sensitizing gene and likely results from the ability of C16 and mivebresib to inhibit distinct sets of genes connected to translation, the impact of which is to enhance p53 induction. BET bromodomain inhibitors have struggled somewhat in clinical trials due to dose-limiting toxicities (*Shorstova et al., 2021*), but their combination with WINi could form the basis of a more effective therapy with less side effects. Moreover, given the mechanism underlying synergy between C16 and BET inhibitors, we would expect this combination to be effective in other wild-type p53 cancer settings where WINi are active, such as

neuroblastoma (*Bryan et al., 2020*) and rhabdoid tumors (*Florian et al., 2022*). Expanded synergy screening is needed to identify and understand the full spectrum of combination approaches that could be used to ultimately enhance and extend the clinical utility of WINi.

## Materials and methods

### Key resources
All key resources are provided in Appendix 1—key resources table.

### Materials availability
Plasmids and cell lines generated in this study are available upon request from the corresponding author (william.p.tansey@vanderbilt.edu).

### Cell lines
MV4;11 (RRID:CVCL_0064), MOLM13 (RRID:CVCL_2119), and K562 (RRID:CVCL_0004) cell lines and their derivatives were cultured in RPMI-1640 media with 10% FBS, 10 U/mL penicillin, and 10 µg/mL streptomycin at 37°C and 5% $CO_2$. HEK293T (RRID:CVCL_1926) cells were cultured in DMEM media with 10% FBS, 10 U/mL penicillin, and 10 µg/mL streptomycin at 37°C and 5% $CO_2$. MV4;11 and MOLM13 cell lines are male. K562 and HEK293T cell lines are female. Cell lines were split every 2–4 d and suspension cells maintained between $1 \times 10^5$ and $1 \times 10^6$ cells/mL. Cell line identity was authenticated by STR profiling. All cell lines tested negative for mycoplasma.

### Generation of *RPL22*-null cell lines
MV4;11, MOLM13, and K562 control (NT) and *RPL22* knockout (KO) cell lines were generated by CRISPR using the multi-guide Synthego Gene Knockout System. Briefly, ribonucleoprotein (RNP) complexes containing Cas9-2NLS (Synthego) and either non-targeting (NT) control sgRNA#1 (Synthego) or RPL22 sgRNAs (Synthego Gene Knockout Kit v2 – human – RPL22) were formed by incubating 90 pmol sgRNA and 10 pmol Cas9-2NLS in Buffer R (Component of Neon Transfection System Kit; Thermo Scientific) at room temperature (RT) for 10 min. MV4;11, MOLM13, or K562 cells were electroporated ($2 \times 10^5$ cells per reaction) with RNP complexes using the Neon Transfection System (Thermo Fisher Scientific) with the following parameters using Buffer R in 10 µL reactions: MV4;11 cells: 1175 V pulse, 40 ms pulse width, one pulse; MOLM13 cells: 1075 V pulse, 30 ms pulse width, two pulses; K562 cells: 1450 V pulse, 10 ms pulse width, three pulses. Cells recovered undisturbed in media absent of antibiotics for 48 hr before expansion and screening for loss of RPL22 expression by western blot analysis.

### Multiplex gene expression assays
Cells were treated with 0.1% DMSO or varying concentrations of C6 or C16 for 24 hr. A custom QuantiGene Plex panel (Thermo Fisher Scientific) was used in conjunction with the QuantiGene Sample Processing Kit for cultured cells (Thermo Fisher Scientific), and QuantiGene Plex Assay kit (Thermo Fisher Scientific) to quantify transcripts following the manufacturer's instructions. Probe regions and accession numbers are as follows: *RPS24* (NM_001026, region 5-334), *RPL35* (NM_007209, region 2-430), *RPL26* (NM_000987, region 37-445), *RPS14* (NM_005617, region 61-552), *RPL32* (NM_000994, region 95-677), *RPS11* (NM_001015, region 139-634), *RPL14* (NM_003973, region 108-530), and *GAPDH* (NM_002046, region 2-407). The average net mean fluorescence intensity was read on a Luminex FLEXMAP 3D System (Invitrogen). Signals from RPGs were normalized internally to those from *GAPDH*, and then to the DMSO control. Dose–response curves from the mean of biological replicates were calculated with the R package *drc* (*Ritz et al., 2015*).

### Western blot analysis
Cells were collected by centrifugation and washed once with ice-cold PBS. Cells were lysed in either RIPA buffer (50 mM Tris, pH 8.0; 150 mM NaCl; 5 mM EDTA; 1.0% NP-40; 0.5% sodium deoxycholate; 0.1% SDS) or Triton-X buffer (50 mM Tris, pH 8.0; 150 mM NaCl; 5 mM EDTA; 1% Triton X-100), each supplemented with protease and phosphatase inhibitors (2× cOmplete, EDTA-free, Protease Inhibitor Cocktail [Roche]; 1× PhosSTOP Phosphatase Inhibitor [Roche]; 100 µg/mL

Pefabloc SC [Roche]), while incubating on ice for 10 min. Chromatin was sheared by brief sonication at 25% on ice, insoluble material cleared by centrifugation, and protein quantified by Pierce BCA Protein Assay (Thermo Scientific). Protein samples were diluted to equal concentrations in lysis buffer and boiled for 5 min in 1× Laemmli Sample Buffer. Samples were run on 4–20% TGX Precast Polyacrylamide Gels (Bio-Rad) or hand-cast single percentage polyacrylamide gels, wet transferred to Amersham Protran Western Blotting Nitrocellulose Membrane (Cytiva) for 1 hr at 100 V in Towbin Buffer (25 mM Tris; 192 mM glycine; 10% methanol), and blocked in 5% milk in TBS-T before incubation overnight with one of the following primary antibodies: anti-p53 (Santa Cruz Biotechnology, Cat# sc-126), anti-RPL22 (Santa Cruz Biotechnology, Cat# sc136413), anti-RPL22L1 (Thermo Fisher Scientific, Cat# PA5-63266), anti-MDM4 (Sigma-Aldrich, Cat# M0445), anti-p21 (Cell Signaling Technology, Cat# 2947), anti-α-actinin (Cell Signaling Technology, Cat# 12413), or anti-GAPDH (Cell Signaling Technology, Cat# 8884). Membranes were washed three times with TBS-T and, if required, incubated with anti-mouse-HRP secondary antibody (Jackson ImmunoResearch Laboratories, Inc, Cat# 115-035-174) or anti-rabbit-HRP (Cell Signaling Technology, Cat# 7074) for 1 hr. Blots were developed with Clarity ECL Western Blotting Substrate (Bio-Rad) and imaged on a ChemiDoc Imaging System (Bio-Rad).

## Immunohistochemistry

MV4;11 cells were treated with 0.1% DMSO (vehicle) or C16 (100 nM) for up to 72 hr or actinomycin D (5 nM) for 6 hr. Cells were fixed in 4% paraformaldehyde (PFA) for 10 min, washed three times with PBS, then cytospun onto slides. Cells were permeabilized with 0.5% Triton X-100 in PBS (PBSTx) for 15 min then blocked with 1% bovine serum albumin in PBSTx (blocking buffer) and immunostained with antibodies against NPM1 (Abcam, ab10530) and gH2A.X pSer139 (Cell Signaling Technologies, 9718). Cells were washed with PBSTx then stained with secondary antibodies (Thermo Fisher, A11001 and A11037). Following PBSTx washes, cells were counterstained with Hoechst (Thermo Fisher, H3570), washed with PBS, then mounted with ProLong Antifade Gold (Thermo Fisher). Images were acquired on using a Plan Fluor ×40 Oil DIC H N2 (NA 1.3, WD 240 mm) objective on a Nikon Ti-2 microscope with a Nikon D-LEDI light source and a Prime BSI Express Scientific sCMOS camera in the Vanderbilt University Cell Imaging Shared Resource. Images were processed and analyzed using NIS-Elements (version 5.42.03) and FIJI (version 2.3.0/1.53q). Images presented are single z-sections of representative cells. To quantify nucleolar localization of NPM1, masks of nuclei were generated from Hoechst channel and nucleolar NPM1 was manually thresholded. The integrated fluorescence intensity of nucleolar NPM1 was then divided by total nuclear NPM1. p-Values were calculated by Student's t-tests comparing treatment samples to DMSO samples within each timepoint.

## Protein synthesis assays

Bulk protein synthesis was measured using the OP-PURO labeling method (*Liu et al., 2012*). MV4;11 cells were treated with either 0.1% DMSO, 2 µM C6, or 100 nM C16 for 24, 48, or 96 hr. For a positive control for inhibition of protein synthesis, MV4;11 cells were treated with 100 µg/ mL cycloheximide (Research Products International) for 30 min. Following treatments, $2 × 10^6$ cells were pulsed with 50 µM O-propargyl-puromycin (Invitrogen, Cat# C10459), or 0.1% DMSO for the 'No OPP' unlabeled control, for 1 hr at 37°C. Cells were collected, washed with ice-cold PBS, and cross-linked in 500 µL Cross-Linking Buffer (1× PBS, 1% formaldehyde) for 15 min on ice. Cross-linked cells were washed with ice-cold PBS and permeabilized in 500 µL Permeabilization Buffer (1× PBS, 3% FBS, 10% saponin) for 5 min at RT. Click-iT reactions containing cells in 500 µL Click-iT Reaction Cocktail (Invitrogen) with 5 µM Alexa Fluor 647 Azide (Invitrogen) were performed for 30 min at RT while protected from light. Cells were washed with PBS + 3% FBS and resuspended in PBS. Fluorescence intensities from Alexa Fluor 647 were measured using a BD LSRFortessa Cell Analyzer and geometric means calculated from 10,000 cells per sample with FlowJo software (BD Bioscience). Experiments were repeated in biological triplicate. Normalized fluorescence values were calculated by setting fluorescence from cycloheximide-treated samples as the baseline. p-Values were calculated by Student's t-tests comparing treatment samples to DMSO samples within each timepoint. Flow Cytometry experiments were performed in the VUMC Flow Cytometry Shared Resource.

## In-gel fluorescence assays for metabolically labeled rRNA

Metabolic labeling of rRNA was performed as previously described, with some modifications (*Wang et al., 2020*). MV4;11 cells were treated with either 0.1% DMSO, 2 µM C6, or 100 nM C16 for 24, 48, or 96 hr. For a positive control treatment for inhibited rRNA transcription, MV4;11 cells were treated with 5 nM actinomycin D (Cayman Chemical Company) for 60 min. Following treatment, cultures were pulsed with 1 mM 2'-azido-2'-deoxycytidine (Biosynth), or 0.1% DMSO for an unlabeled control, for 12 hr and total RNA isolated using TRIzol Reagent (Invitrogen) as per the manufacturer's instructions. Pelleted RNA was resuspended in 20 µL nuclease-free water and treated with DNase I (New England BioLabs) for 10 min at 37°C in the presence of RNasin Ribonuclease Inhibitor (Promega). DNase-treated RNA was purified with the RNA Clean and Concentrator-25 Kit (Zymo Research) as per the manufacturer's instructions. SPAAC reactions containing 12.5 µg RNA, 100 µM MB 680R DBCO (Vector Laboratories), 20 U RNasin Ribonuclease Inhibitor (Promega), and 1× PBS were incubated for 2 hr at 37°C, and then RNA purified with the RNA Clean and Concentrator-25 Kit (Zymo Research) as per the manufacturer's instructions. RNA was subjected to electrophoretic separation on 1% TAE-agarose gels and MB 680R-labeled RNA imaged on an Odyssey CLx Imager (LI-COR). Total RNA was stained with SYBR Safe DNA Gel Stain (Invitrogen) and imaged on a ChemiDoc Imaging System (Bio-Rad). Fluorescence signals from 28S and 18S bands were quantified using Empiria Studio (LI-COR).

## Ribo-seq

Ribo-seq was performed as previously described with some modifications (*McGlincy and Ingolia, 2017*). MV4;11 cells treated for 48 hr with either 0.1% DMSO, 2 µM C6, or 100 nM C16 were washed with ice-cold PBS and lysed in 400 µL Lysis Buffer (20 mM Tris, pH 7.4; 150 mM NaCl; 15 mM MgCl$_2$; 1 mM DTT; 100 µg/mL cycloheximide; 1% Triton X-100; 25 U/mL Turbo DNase I) by incubation on ice for 10 min followed by homogenization by syringe. Lysates were cleared by centrifugation at 4°C, and RNA quantified by Qubit RNA HS Assay (Invitrogen) following the manufacturer's instructions. 30 µg RNA was diluted in 200 µL Polysome Buffer (20 mM Tris, pH 7.4; 150 mM NaCl; 5 mM MgCl$_2$; 1 mM DTT; 100 µg/mL cycloheximide) and incubated with 15 U RNase I (Lucigen) for 45 min while rotating at RT. RNA digestion was quenched with 10 µL SUPERaseIn RNase Inhibitor (Invitrogen) and samples transferred to 13 mm × 51 mm ultracentrifuge tubes (Beckman-Coulter), underlaid with 900 µL 1 M sucrose in polysome buffer supplemented with 20 U/mL SUPERaseIn RNase Inhibitor, and centrifuged at 540,628 × *g* 1 hr at 4°C. Ribosome pellets were suspended in TRIzol Reagent (Invitrogen), and RNA extracted from ribosome pellets by Direct-zol RNA MiniPrep Kit (Zymo Research). RNA and carrier glycogen were precipitated by adding 1.5 volumes 100% isopropanol supplemented with 0.12 M NaOAc, pH 5.5, followed by incubation on dry ice for 30 min and centrifugation at 16,800 × *g* for 30 min at 4°C. RNA was resuspended in 5 µL 10 mM Tris, pH 8.0, and 1× Denaturing Sample Loading Buffer (98% formamide; 10 mM EDTA; 300 µg/mL bromophenol blue) and subjected to electrophoresis on 15% polyacrylamide TBE-Urea gels (Invitrogen). Gels were stained briefly with 1× SYBR Gold (Invitrogen), 17–34 nucleotide fragments excised, and RNA fragments extracted by mechanical disruption, suspension in 500 µL RNA Gel Extraction Buffer (300 mM NaOAc, pH 5.5; 1 mM EDTA; 0.25% SDS), freezing on dry ice for 30 min, and rotating overnight at RT. Polyacrylamide was removed by centrifugation through Costar Spin-X columns (Corning) and RNA precipitated with isopropanol as described above. RNA was dephosphorylated by incubation with 5 U T4 Polynucleotide Kinase in 1× T4 PNK Buffer (New England BioLabs) supplemented with SUPERaseIn RNase Inhibitor for 1 hr at 37°C and ligated to bar-coded linkers (NI-810: /5Phos/NNNNNATCGTAGATCGGAAGAGCACACGTCTGAA/3ddC/; NI-811: /5Phos/NNNNNAGCTAAGATCGGAAGAGCACACGTCTGAA/3ddC/; NI-812: /5Phos/NNNNNCGTAAAGATCGGAAGAGCACACGTCTGAA/3ddC/) pre-adenylated with 100 U of T4 RNA Ligase 2, truncated K227Q, in 1X T4 RNA Ligase Buffer (New England BioLabs) supplemented with 35% w/v PEG-8000 and incubated at 37°C for 3 hr. Ligation was verified by electrophoresis, samples combined, and linker-ligated RNA precipitated with isopropanol as described above. Ribosomal RNA was depleted from samples using the RiboCop rRNA Depletion Kit (Lexogen) and RNA precipitated with isopropanol as described above. Linker-ligated RNA was reverse transcribed with 200 U SuperScript III in 1× First Strand Buffer (Invitrogen), dNTPs, DTT, 10 U SUPERaseIn RNase Inhibitor and primer NI-802 (/5Phos/NNAGATCGGAAGAGCGTCGTGTAGGGAAAGAG/iSp18/GTGACTGGAGTTCAGACGTGTGCTC). RNA template was hydrolyzed for 15 min at 98°C in the presence of 0.1 M NaOH and cDNA precipitated with isopropanol as described previously.

Reverse-transcribed DNA was subjected to electrophoresis on 15% polyacrylamide Novex TBE-Urea gels (Invitrogen) and the 105-nucleotide reverse-transcription product excised from polyacrylamide as described above, except with DNA Gel Extraction Buffer (300 mM NaCl; 10 mM Tris, pH 8.0; 1 mM EDTA). cDNA was circularized with 100 U CircLigase ssDNA Ligase (Lucigen) in the presence of 1× CircLigase Buffer, ATP, and $MnCl_2$ at 60°C for 1 hr followed by 80°C for 10 min. Circularized cDNA was quantified by qPCR, amplified using Phusion Polymerase (New England BioLabs) with unique dual-indexed primers (UDI0050_i5: AATGATACGGCGACCACCGAGATCTACACGCTCCGACACACTCT TTCCCTACACGACGCTCTTCCGATCT; UDI0050_i7: CAAGCAGAAGACGGCATACGAGATTAGAGC GCGTGACTGGAGTTCAGACGTGT; UDI0051_i5: AATGATACGGCGACCACCGAGATCTACACATA CCAAGACACTCTTTCCCTACACGACGCTCTTCCGATCT; UDI0051_i7: CAAGCAGAAGACGGCA TACGAGATAACCTGTTGTGACTGGAGTTCAGACGTGT; UDI0052_i5: AATGATACGGCGACCACCGA GATCTACACGCGTTGGAACACTCTTTCCCTACACGACGCTCTTCCGATCT; UDI0052_i7: CAAG CAGAAGACGGCATACGAGATGGTTCACCGTGACTGGAGTTCAGACGTGT), amplicons subjected to electrophoresis on 8% polyacrylamide Novex gels (Invitrogen), and products >160 bp excised as described. Libraries were submitted to VANTAGE (Vanderbilt Technologies for Advanced Genomics) for sequencing on a NovaSeq 6000.

## Ribo-seq data analysis

Adapters were trimmed from reads using *cutadapt* (*Martin, 2011*), and UMIs removed from reads and attached to read IDs using *UMI-tools* (*Smith et al., 2017*). Reads were demultiplexed using *sabre* and aligned against ribosomal RNA using *bowtie2* (*Langmead and Salzberg, 2012*). Reads not mapping to rRNA were mapped to the hg19 transcriptome using *STAR* (*Dobin et al., 2013*) and deduplicated using *UMI-tools* (*Smith et al., 2017*). Count tables for reads mapping to central ORFs were generated using the coverage command from *bedtools* (*Quinlan and Hall, 2010*). After batch removal, Ribo-seq read counts were normalized to mRNA read counts using *Xtail* (*Xiao et al., 2016*) to calculate translation efficiencies and statistics. FDR values were calculated using the Cochran–Mantel–Haenszel test (CMH). Genes with significantly altered translation efficiencies were those with FDR < 0.05 and absolute log2FC > 0.25. Identification of optimal RPF P-site offsets, RPF triplet periodicity, and RPF localization to CDS and UTR regions was performed with the R package *riboWaltz* (*Lauria et al., 2018*).

## RNA-seq

For RNA-seq performed in parallel with Ribo-seq, RNA was isolated by Direct-zol RNA MiniPrep Kit (Zymo Research) from 100 µL of cell lysates after homogenization by syringe and clearing by centrifugation. For combination WINi/BETi treatment, MV4;11 cells were treated for 48 hr with either 0.2% DMSO, 100 nM C16, 2.5 nM mivebresib, or combined 100 nM C16 and 2.5 nM mivebresib, and RNA isolated by Direct-zol RNA MiniPrep Kit (Zymo Research) with on-column DNAse-treatment. For RNA-seq in MV4;11 NT and RPL22 KO cells, cultures were treated for 48 hr with either 0.1% DMSO or 100 nM C16 before RNA isolation as described above for WINi/BETi RNA-seq. For all RNA-seq experiments, RNA was submitted to the Vanderbilt Technologies for Advanced Genomics (VANTAGE) core facility for library preparation with rRNA-depletion using standard Illumina protocols and sequencing on an Illumina NovaSeq 6000.

## RNA-seq data analysis

Adapters were trimmed from RNA-seq reads using *cutadapt* (*Martin, 2011*) and reads aligned to the hg19 genome using *STAR* (*Dobin et al., 2013*). Gene expression was quantified using *featureCounts* (*Liao et al., 2014*) and differential analysis performed using *DESeq2* (*Love et al., 2014*) which calculates p-values through the Wald test and adjusts p-values by the Benjamini–Hochberg procedure to calculate FDR. Changes in levels of alternative splicing events were quantified using *rMATS,* which calculates changes in exon inclusion levels ($\Delta\psi$), and p-values through a likelihood-ratio test. Genes with significantly altered transcript levels are those with FDR < 0.05. Significant changes in alternative splicing events are those with FDR < 0.05 and $\Delta\psi$ > 5%.

## Generation of Cas9-expressing MV4;11 cells

To generate Cas9 expression lentivirus, HEK293T cells were transfected with the viral transfer plasmid lentiCas9-Blast (*Sanjana et al., 2014*) (gift from Feng Zhang; Addgene plasmid # 52962), the viral

packaging plasmid psPAX2 (gift from Didier Trono; Addgene plasmid # 12260), and the viral envelope plasmid pMD2.G (gift from Didier Trono; Addgene plasmid # 12259) using Lipofectamine 3000 Transfection Reagent (Invitrogen). After 48 hr, virus-containing media was collected and used to transduce MV4;11 cells by spinfection (2 hr; 1000 × $g$; RT; 8 µg/mL hexadimethrine bromide). Following spinfection, virus-containing media was replaced with fresh media and cells allowed to recover for 48 hr before selection with 10 µg/mL blasticidin (Research Products International). A clonal MV4;11 Cas9 cell line was established by serial dilution of the population and screening for retention of WINi sensitivity.

## Tier 1 CRISPR screen

Tier 1 CRISPR screens were performed essentially as described (*Joung et al., 2017*). Briefly, the Human GeCKOv2 CRISPR Knockout Pooled Library (A+B) in the lentiGuide-Puro vector backbone (gift from Feng Zhang; Addgene plasmid # 1000000048) was amplified and purified as directed by Addgene. Lentiviral particles were generated by transfecting HEK293T cells with the GeCKOv2 CRISPR Knockout Pooled Plasmid Library, psPAX2 (gift from Didier Trono; Addgene plasmid # 12260), and pMD2.G (gift from Didier Trono; Addgene plasmid # 12259) using Lipofectamine 3000 Transfection Reagent (Invitrogen). After 48 hr, viral media was collected, aliquoted, and stored at –80°C. In duplicate, clonal Cas9-expressing MV4;11 cells were transduced by spinfection (2 hr; 1000 × $g$; RT; 8 µg/mL hexadimethrine bromide) with a volume of virus-containing media sufficient to infect 30% of cells and at a scale to generate >200 transduced cells per sgRNA in the library. Cells recovered in fresh media overnight, were split 1:2, and selected with 1 µg/mL puromycin for 48 hr to generate the MV4;11 Cas9+GeCKOv2 population.

MV4;11 Cas9 and MV4;11 Cas9+GeCKOv2 cells were treated with either 0.1% DMSO or 2 µM C6, replenished every 3 d with fresh media and C6, and counted daily by trypan blue exclusion. DMSO-treated populations were grown until cultures reached >8 × 10$^5$ cells/mL to verify C6-treatment efficacy. C6-treated MV4;11 Cas9+GeCKOv2 populations were maintained below 8 × 10$^5$ cells/mL and grown until a resistant population emerged relative to C6-treated MV4;11 Cas9 cells. Genomic DNA was isolated from MV4;11 Cas9+GeCKOv2 cells collected before and following sustained C6 treatment using the Quick-gDNA MidiPrep Kit (Zymo Research) as per the manufacturer's directions. Sequencing libraries were generated by amplifying sgRNA sequences from genomic DNA using barcoded Illumina-compatible adapter-containing primers and NEBNext High-Fidelity 2× PCR Master Mix (New England BioLabs). PCR products were pooled and purified with a ZymoSpin V column with Reservoir (Zymo Research). Libraries were sequenced on an Illumina NextSeq 500 in the Vanderbilt Technologies for Advanced Genomics (VANTAGE) core facility.

## Cloning targeted sgRNA library for second-tier screen

The tier 2 sgRNA plasmid library was generated as previously described with some modifications (*Joung et al., 2017*). Briefly, sgRNA sequences against a curated collection of genes and 200 non-targeting control sgRNA sequences were extracted from the Brunello sgRNA Library (*Doench et al., 2016*). For genes of interest not included in the Brunello library, four sgRNAs targeting each gene were designed with the CHOPCHOP sgRNA design tool (*Labun et al., 2019*). sgRNA sequences were appended with 5′ and 3′ flanking sequences and synthesized as an Oligo Pool (*Figure 4—source data 2*; Twist Bioscience) followed by PCR amplification using NEBNext HiFidelity 2× Master Mix (New England BioLabs) with Fwd primer (GTAACTTGAAAGTATTTCGATTTC TTGGCTTTATATATCTTGTGGAAAGGACGAAACACC) and KO Rev primer (ACTTTTTCAAGTTGAT AACGGACTAGCCTTATTTTAACTTGCTATTTCTAGCTCTAAAAC). PCR amplicons were subjected to agarose gel size selection using the NucleoSpin Gel and PCR Clean-up Kit (Macherey-Nagel). Amplicons were cloned into BsmBIv2-digested (New England BioLabs) lentiGuide-PURO plasmid (gift from Feng Zhang; Addgene plasmid # 52963) via Gibson Assembly (New England BioLabs). Gibson Assembly products were precipitated with isopropanol and electroporated into Endura ElectroCompetent *Escherichia coli* (Lexogen). Amplified plasmids were isolated from *E. coli* using the Nucleobond Xtra Maxi EF Kit (Macherey-Nagel) and adequate representation of sgRNAs in the library was verified by next-generation sequencing and analysis with the Python script *count_spacers.py* (*Joung et al., 2017*).

## Tier 2 CRISPR screen

Tier 2 sgRNA Library lentiviral particles were generated and MV4;11 Cas9 cells transduced as described above for the tier 1 screen at a scale to achieve >500 cells per sgRNA in the library. MV4;11 Cas9+Targeted sgRNA Library populations were treated with either 0.1% DMSO, 2 µM C6, or 100 nM C16 for 15 d. Cultures were maintained below $8 \times 10^5$ cells/mL and cultures replenished every 3 d with media and fresh DMSO, C6, or C16. Genomic DNA was isolated, and Illumina-compatible next-generation sequencing libraries generated as described above for the tier 1 screen. Libraries were sequenced on an Illumina NovaSeq 6000 in the Vanderbilt Technologies for Advanced Genomics (VANTAGE) core facility.

## CRISPR screen data analysis

Adapters were trimmed from reads using *cutadapt* (*Martin, 2011*). Generation of sgRNA count tables and determination of significant gene-level alterations in sgRNA representation were performed using *MAGeCK* (*Li et al., 2014*), which utilizes a negative binomial model to determine p-values of sgRNA changes and ranks sgRNAs by significance. Gene-level alterations and p-values were calculated from the ranked list of sgRNAs using the modified robust ranking aggregation (α-RRA) algorithm and FDR values calculated by the Benjamini–Hochberg procedure. Tier 1 screen analysis compared populations before and after C6 treatment. Tier 2 screen analysis compared DMSO-treated populations to C6- or C16-treated populations. Significantly enriched or depleted genes were those with FDR <0.05.

## Cell viability assays

Opaque 384-well plates were seeded with 250 cells per well in 25 µL media supplemented with either 0.1% DMSO or a threefold dilution series of C6 or C16, all in technical quadruplicate wells. Cells were grown for 72 hr before equilibrating to RT and addition of 12.5 µL CellTiter-Glo Luminescent Cell Viability Assay reagent (Promega). At RT and protected from light, plates were rocked for 5 min, incubated for 20 min, and luminescence measured on a GloMax Explorer Multimode Microplate Reader (Promega). To calculate relative cell viability, mean fluorescence from quadruplicate treatment wells was divided by mean fluorescence from quadruplicate DMSO wells. Dose–response curves, $GI_{50}$ concentrations, and standard error values were calculated from at least three biological replicates with the R package *drc* (*Ritz et al., 2015*).

## Synergy assays

Opaque 384-well plates were seeded with 250 cells per well in 25 µL media supplemented with either 0.2% DMSO, a threefold dilution series of either C16 or compound 2, or a combination of threefold dilutions of both C16 and compound 2 covering a $7 \times 7$ dose matrix, all in quadruplicate wells. Compound 2 consisted of either nutlin-3a (Cayman Chemical Company), rapamycin (MedChem Express), pinometostat (Cayman Chemical Company), harmine (Sigma-Aldrich), mivebresib (Cayman Chemical Company), venetoclax (Cayman Chemical Company), etoposide (Cayman Chemical Company), olaparib (Cayman Chemical Company), VE-821 (Cayman Chemical Company), pemrametostat (Selleck Chemicals), or alvespimycin (Cayman Chemical Company). Following 72 hr, plates were equilibrated to room temperature and 12.5 µL CellTiter-Glo Cell Viability Assay (Promega) reagent added to each well. While protected from light, plates were rocked for 5 min, incubated for 20 min, and luminescence measured on a GloMax Explorer Multimode Microplate Reader (Promega).

## Synergy assay data analysis

Technical replicate wells were averaged and resulting means used to calculate relative cell viability by dividing drug treatment by DMSO treatment. Mean δ-scores and standard deviations were calculated from three biological replicates via *SynergyFinder Plus* (*Zheng et al., 2022*) using the ZIP model (*Yadav et al., 2015*). ZIP δ-scores represent the percent of growth inhibition beyond that expected if the agents do not potentiate one another. δ-scores greater than zero are synergistic, δ-scores of zero are additive, and δ-scores less than zero are antagonistic. Statistical significances of peak synergistic and antagonistic δ-scores were calculated by one-sample *t*-tests using the *tsum.test* function from the R package *PASWR*. Significant synergy and antagonism δ-scores were those with p<0.05.

## Quantitative proteomics

In quadruplicate, MV4;11 cells were seeded at $2 \times 10^5$ cells/mL and treated with 0.1% DMSO or 250 nM C16 for either 24 or 72 hr. Cells were collected by centrifugation and washed three times

with ice-cold 1× PBS before lysis on ice in SDS Lysis Buffer (5% SDS; 50 mM ammonium bicarbonate). Chromatin was sheared by brief sonication at 25% on ice and insoluble material cleared by centrifugation. Soluble proteins were quantified by Pierce BCA Protein Assay (Thermo Scientific). Of note, protein was isolated from equivalent cell numbers at 24 hr as changes in proliferation are not observed until beyond 24 hr WINi treatment in MV4;11 cells. At 24 hr, DMSO-treated cultures yielded 344.75 ± 21.7 µg total soluble protein and C16-treated cultures yielded 366.50 ± 15.8 µg total soluble protein (mean ± SEM).

Protein samples for LC-MS/MS analyses were prepared by S-Trap (ProtiFi) digestion. Protein samples (50 µg) were reduced with DTT (MilliporeSigma) at a final concentration of 20 mM at 95°C for 10 min and alkylated with iodoacetamide (MilliporeSigma) at a final concentration of 40 mM at RT for 30 min in the dark. Aqueous phosphoric acid (Fisher Scientific) was added to the samples at a final concentration of 1.2% followed by 90% methanol containing 100 mM TEAB at 6.6 times the volume of the sample. The samples were loaded on the S-Trap micro columns and centrifuged at 4000 × $g$ until all the volume was passed through the column. The columns were washed four times with 150 µL 90% methanol containing 100 mM TEAB, pH 7.1. Proteins were digested with trypsin gold (Promega) at 1:50 enzyme to protein ratio in 50 mM TEAB, pH 8.0, for 1 hr at 47°C. Peptides were eluted by serial addition of 40 µL each of 50 mM TEAB, 0.2% formic acid, and 35 µL of 0.2% formic acid in 50% acetonitrile. Eluted peptides were dried in a speed-vac concentrator, resuspended in aqueous 0.1% formic acid, and analyzed by LC-coupled tandem mass spectrometry (LC-MS/MS).

An analytical column (360 µm O.D. × 100 µm I.D.) was packed with 25 cm of C18 reverse-phase material (Jupiter, 3 µm beads, 300 Å; Phenomenex) directly into a laser-pulled emitter tip. Peptides were loaded on the reverse phase column using a Dionex Ultimate 3000 nanoLC and autosampler. The mobile phase solvents consisted of 0.1% formic acid, 99.9% water (solvent A) and 0.1% formic acid, 99.9% acetonitrile (solvent B). Peptides were gradient eluted at a flow rate of 350 nL/min using a 120 min gradient. The gradient consisted of the following: 1–100 min, 2–38% B; 100–108 min, 38–90% B; 108–110 min, 90% B; 110–111 min, 90–2% B; 111–120 min (column re-equilibration), 2% B. Upon gradient elution, peptides were analyzed using a data-dependent method on an Orbitrap Exploris 480 mass spectrometer (Thermo Scientific), equipped with a nanoelectrospray ionization source. The instrument method consisted of MS1 using an MS AGC target value of $3 \times 10^6$, followed by up to 15 MS/MS scans of the most abundant ions detected in the preceding MS scan. The intensity threshold for triggering data-dependent scans was set to $1 \times 10^4$, the MS2 AGC target was set to $1 \times 10^5$, dynamic exclusion was set to 20 s, and HCD collision energy was set to 30 nce.

## Quantitative proteomics data analysis

For identification of peptides, LC-MS/MS data were searched with *Maxquant,* version 2.0.1.0 (*Cox and Mann, 2008*). MS/MS spectra were searched with the *Andromeda* search engine (*Cox et al., 2011*) against a human database created from the UniprotKB protein database (*Bateman et al., 2021*) and the default *Maxquant* contaminants. Default parameters were used for *Maxquant,* with the addition of selecting LFQ and match between runs as a global parameter. *Maxquant* parameters included first and main search mass tolerances of 20 ppm and 4.5 ppm, respectively. Variable modifications included methionine oxidation and N-terminal acetylation, and carbamidomethyl cysteine was selected as a fixed modification. A maximum of two missed cleavages was allowed. The false discovery rate (FDR) was set to 0.01 for peptide and protein identifications. Label-free quantitative (LFQ) analysis of identified proteins was performed with the *MSstats* R package (*Choi et al., 2014*), version 4.0.1, using default parameters, which include the following: equalize medians for the normalization method, log2 transformation, Tukey's median polish as the summary method, and model-based imputation. Protein fold changes were considered as significant with adjusted p-values ≤0.05.

## RNA isolation and cDNA synthesis

Cell pellets were suspended in TRIzol Reagent (Invitrogen), rotated for 15 min at RT, and insoluble cellular debris pelleted by centrifugation. The soluble fraction was mixed with equal volume of 100% ethanol and RNA was isolated using the Direct-zol RNA Miniprep Kit (Zymo Research) according to the manufacturer's instructions, including on-column DNA digestion. Complementary DNA (cDNA) was synthesized in 20 µL cDNA reactions containing 1 µg RNA, Random Hexamers (Invitrogen), and SuperScript III Reverse Transcriptase (Invitrogen) as per the manufacturer's instructions. Final cDNA

products were diluted fivefold with nuclease-free water before use in semi-quantitative RT-PCR or quantitative RT-PCR.

## Semi-quantitative RT-PCR

PCR reactions were performed with 2 µL cDNA template using primers amplifying splicing variants of *RPL22L1* (RPL22L1_RTPCR_F: ATGGCGCCGCAGAAAGAC; RPL22L1_RTPCR_R: CTAGTCCT CCGACTCTGATT) or *MDM4* (MDM4_RTPCR_F: GAAAGACCCAAGCCCTCT; MDM4_RT_PCR_F: GCAGTGTGGGGATATCGTCT), or within *GAPDH* (GAPDH_RTPCR_F: TCACCAGGGCTGCTTTTAAC; GAPDH_RTPCR_R: ATCGCCCCACTTGATTTTGG) using Taq DNA Polymerase (New England BioLabs) with primer-specific annealing temperatures and cycle numbers (RPL22L1: 50°C, 30 cycles; MDM4: 54°C, 33 cycles; GAPDH: 52°C, 27 cycles). PCR products were electrophoretically separated on 2% agarose gels in TBE buffer, gels incubated 30 min in TBE buffer containing 1× SYBR Safe DNA Stain (Invitrogen) with agitation, and imaged on a ChemiDoc Imaging System (Bio-Rad).

## Quantitative RT-PCR

qPCR reactions containing 1× KAPA SYBR Fast qPCR Master Mix (Roche), transcript-specific primers, and 2 µL cDNA template were performed in technical duplicate wells on a C1000 Touch Thermal Cycler (Bio-Rad) with a CFX96 Touch Real-Time PCR Detection System (Bio-Rad). Primer pairs targeted total *RPL22L1* (RPL22L1ab_qPCR_F: tcgagtggttgcatctgaca; RPL22L1ab_qPCR_R: tcctccgactctgatt catct), *RPL22L1a* (RPL22L1a_qPCR_F: cgccgcagaaagacaggaa; RPL22L1a_qPCR_R: ctcccgtagaaattgc tcaaaat), *RPL22L1b* (RPL22L1b_qPCR_F: cgcagaaagacaggaagcc; RPL22L1b_qPCR_R: tgcaaaactagg gaagagaacc), *MDM4* exon 5–6 junction (MDM4_Jnct_5_6_qPCR_F: AGAATCTTGTCACTTTAGCC ACT; MDM4_Jnct_5_6_qPCR_R: CGAGAGTCTGAGCAGCATCT), *MDM4* exon 6–7 junction (MDM4_ Jnct_6_7_qPCR_F: TCAAGACCAACTGAAGCAAAGT; MDM4_Jnct_6_7_qPCR_R: TAGGCAGTGTGG GGATATCG), *MDM4* exon 4 (MDM4_Ex_4_qPCR_F: AGCAACTTTATGATCAGCAGGAG; MDM4_ Ex_4_qPCR_R: GACGTCCCAGTAGTTCTCCC), *MDM4* exon 7 (MDM4_Ex_7_qPCR_F: AGAGGAAA GTTCCACTTCCAGA; MDM4_Ex_7_qPCR_R: ATGCTCTGAGGTAGGCAGTG), or *GAPDH* (GAPDH_ qPCR_F: AAGGTGAAGGTCGGAGTCAAC; GAPDH_qPCR_R: GTTGAGGTCAATGAAGGGGTC). Ct values for each well were determined by the Bio-Rad CFX Manager Software v3.1 using the regression model, and mean Ct values from technical replicate wells used for subsequent calculations. Relative isoform levels were calculated via the $2^{(-\Delta\Delta Ct)}$ algorithm by internally normalizing isoform-specific Ct values to GAPDH Ct values, then relative to DMSO-treatment.

## Quantification and statistical analysis

The statistical test, threshold for statistical significance, and *n* for each experiment, representing biological replicates, can be found in the figure legends.

## Structure alignment

Images of C6 and C16 bound to the WIN-Site of WDR5 and overlaid structures in WDR5-binding conformations were generated with *PyMOL* using published X-ray crystal structures (C6, PDB: 6E23 [*Aho et al., 2019a*]; C16, PDB: 6UCS [*Tian et al., 2020*]).

## GSEA and ORA

Gene set enrichment analyses (GSEA) and over-representation analyses (ORA) were performed with the R package *fgsea* (*Korotkevich et al., 2021*) using the Molecular Signatures Database v7.4 (*Subramanian et al., 2005*; *Liberzon et al., 2011*; *Liberzon et al., 2015*). Significantly enriched or depleted gene sets were those with FDR < 0.05.

# Acknowledgements

For reagents, we thank D Trono and F Zhang. For assistance, we thank Lu Chen, David Cortez, Rachel Green, Matthew Hall, Ian Macara, Kavi Mehta, Bill Moore, Jonathan Shrimp, and Jamie Wangen. The VANTAGE Shared Resource is supported by the CTSA Grant (RR024975), the Vanderbilt Ingram Cancer Center (CA068485), the Vanderbilt Vision Center (EY008126), and NIH/NCRR (RR030956). Core services for QuantiGene assays performed through Vanderbilt University Medical Center's Digestive Disease Research Center were supported by NIH grant DK058404. The VUMC Flow Cytometry Shared

Resource is supported by the Vanderbilt Cell Imaging Shared Resource and the Vanderbilt Ingram Cancer Center and the Vanderbilt Digestive Disease Research Center. We acknowledge support of the Vanderbilt Proteomics Core in the Mass Spectrometry Research Center, supported in part by the Vanderbilt Ingram Cancer Center. This work was supported by awards from the NIH/NCI—under Chemical Biology Consortium Contract No. HHSN261200800001E (SWF and WPT), and CA200709 (WPT)—as well as grants from the Robert J Kleberg, Jr., and Helen C Kleberg Foundation (WPT and SWF). BCG was supported by the Brock Family Fellowship, the NCI (CA217834/CA268703), and an American Society for Clinical Oncology Young Investigator's Award.

## Additional information

### Competing interests

Michael R Savona: Receives research funding from ALX Oncology, Astex, Incyte, Takeda and TG Therapeutics; has stock in Karyopharm and Ryvu; serves on advisory boards or consults for BMS, CTI, Forma, Geron, GSK, Karyopharm, Rigel, Ryvu, Taiho and Treadwell. Taekyu Lee, Stephen Fesik: Patents: Lee T, Alvarado J, Tian J, Meyers KM, Han C, Mills JJ, Teuscher KB, Stauffer SR, Fesik SW. WDR5 inhibitors and modulators. WO 2020086857. 30 April 2020; Lee T, Han C, Mills JJ, Teuscher KB, Tian J, Meyers KM, Chowdhury S, Fesik SW. WDR5 inhibitors and modulators. WO 2020247679. 10 December 2020; Lee T, Teuscher KB, Tian J, Meyers KM, Chowdhury S, Fesik SW. WDR5 Inhibitors and modulators. WO 2021092525. 14 May 2021; Lee T, Teuscher KB, Chowdhury S, Tian J, Meyers KM, Fesik SW. WDR5 Inhibitors and modulators. WO2022236101. 10 November 2022. William P Tansey: Patents: Fesik SW, Stauffer SR, Salovich JM, Tansey WP, Wang F, Phan J, Olejniczak ET, inventors. WDR5 inhibitors and modulators. United States Patent US 10,501,466. 10 December 2019; Fesik SW, Stauffer SR, Tansey WP, Olejniczak ET, Phan J, Wang F, Jeon K, Gogliotti RD, inventors. WDR5 inhibitors and modulators. United States Patent US 10,160,763. 25 December 2018. The other authors declare that no competing interests exist.

### Funding

| Funder | Grant reference number | Author |
|---|---|---|
| National Institutes of Health | HHSN261200800001E | Stephen Fesik<br>William P Tansey |
| National Institutes of Health | CA200709 | William P Tansey |
| Robert J. Kleberg, Jr. and Helen C. Kleberg Foundation | | Stephen Fesik<br>William P Tansey |
| National Institutes of Health | CA217834 | Brian C Grieb |
| National Institutes of Health | CA268703 | Brian C Grieb |
| American Society for Clinical Oncology | Young Investigator Award | Brian C Grieb |

The funders had no role in study design, data collection and interpretation, or the decision to submit the work for publication.

### Author contributions

Gregory Caleb Howard, Conceptualization, Visualization, Writing – original draft, Investigation, Funding acquisition, Software, Supervision, Formal analysis, Writing – review and editing; Jing Wang, Data curation, Software, Formal analysis, Visualization, Investigation; Kristie L Rose, Visualization, Writing – original draft, Investigation, Funding acquisition, Software, Writing – review and editing; Camden Jones, Purvi Patel, Logan Vlach, Shelly L Lorey, Brian C Grieb, Brianna N Smith, Macey J Slota, Elizabeth M Reynolds, Soumita Goswami, Investigation, Writing – review and editing; Tina Tsui, Qi Liu, Data curation, Software, Formal analysis, Investigation; Andrea C Florian, Investigation,

Funding acquisition, Writing – review and editing; Michael R Savona, Supervision, Investigation; Frank M Mason, Formal analysis, Investigation, Visualization, Writing – review and editing; Taekyu Lee, Visualization, Investigation; Stephen Fesik, Supervision, Funding acquisition, Investigation; William P Tansey, Conceptualization, Supervision, Funding acquisition, Writing – original draft, Investigation

### Author ORCIDs
Gregory Caleb Howard  https://orcid.org/0000-0002-8373-4573
Frank M Mason  https://orcid.org/0000-0003-1338-494X
William P Tansey  https://orcid.org/0000-0002-3900-0978

Reviewer #1 (Public Review): https://doi.org/10.7554/eLife.90683.3.sa1
Reviewer #2 (Public review): https://doi.org/10.7554/eLife.90683.3.sa2
Author response https://doi.org/10.7554/eLife.90683.3.sa3

## Additional files

### Supplementary files
• MDAR checklist

### Data availability
Ribo-Seq, RNA-Seq, and CRISPR screen data are deposited at Gene Expression Omnibus (GEO) with accession number GSE206931. Quantitative proteomics data are deposited at the ProteomeXchange Consortium via the PRIDE partner repository with identifier PXD035129. All these data are publicly available.

The following datasets were generated:

| Author(s) | Year | Dataset title | Dataset URL | Database and Identifier |
| --- | --- | --- | --- | --- |
| Howard GC, Tansey WP, Wang J, Liu Q | 2023 | Multiomic characterization of WDR5 WIN site inhibition reveals actionable synergies for MLL-rearranged leukemia | https://www.ncbi.nlm.nih.gov/geo/query/acc.cgi?acc=GSE206931 | NCBI Gene Expression Omnibus, GSE206931 |
| Rose KL | 2023 | Proteome alterations in MLL-rearranged leukemia cells following WIN Site inhibition | https://proteomecentral.proteomexchange.org/cgi/GetDataset?ID=PXD035129 | ProteomeXchange, PXD035129 |

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

# Appendix 1

**Appendix 1—key resources table**

| Reagent type (species) or resource | Designation | Source or reference | Identifiers | Additional information |
|---|---|---|---|---|
| Gene (*Homo sapiens*) | RPL22 | NA | ENSEMBL:ENSG00000116251 | |
| Gene (*H. sapiens*) | RPL22L1 | NA | ENSEMBL:ENSG00000163584 | |
| Gene (*H. sapiens*) | MDM4 | NA | ENSEMBL:ENSG00000198625 | |
| Strain, strain background (*Escherichia coli*) | Endura ElectroCompetent Cells | Lucigen | Cat# 60242-2 | |
| Cell line (*H. sapiens*) | 'MV4;11' | ATCC | Cat# CRL-9591; RRID:CVCL_0064 | |
| Cell line (*H. sapiens*) | 'MV4;11 NT' | This study | NA | See 'Generation of RPL22-null cell lines' |
| Cell line (*H. sapiens*) | 'MV4;11 RPL22 KO' | This study | NA | See 'Generation of RPL22-null cell lines' |
| Cell line (*H. sapiens*) | 'MV4;11 Cas9' | This study | NA | See 'Generation of Cas9-expressing MV4;11 cells' |
| Cell line (*H. sapiens*) | MOLM13 | DMSZ | Cat# ACC554; RRID:CVCL_2119 | |
| Cell line (*H. sapiens*) | MOLM13 NT | This study | NA | See 'Generation of RPL22-null cell lines' |
| Cell line (*H. sapiens*) | MOLM13 RPL22 KO | This study | NA | See 'Generation of RPL22-null cell lines' |
| Cell line (*H. sapiens*) | K562 | ATCC | Cat# CCL-243; RRID:CVCL_0004 | |
| Cell line (*H. sapiens*) | K562 NT | This study | NA | See 'Generation of RPL22-null cell lines' |
| Cell line (*H. sapiens*) | K562 RPL22 KO | This study | NA | See 'Generation of RPL22-null cell lines' |
| Cell line (*H. sapiens*) | HEK293T | ATCC | Cat# CRL-11268; RRID:CVCL_1926 | |
| Antibody | Anti-p53 (DO-1) (mouse monoclonal) | Santa Cruz Biotechnology | Cat# sc-126; RRID:AB_628082 | (1:2000) |
| Antibody | Anti-RPL22 (52) (mouse monoclonal) | Santa Cruz Biotechnology | Cat# sc-136413; RRID:AB_10658965 | (1:1000) |
| Antibody | Anti-RPL22L1 (rabbit polyclonal) | Thermo Fisher Scientific | Cat# PA5-63266; RRID:AB_2646731 | (1:1000) |
| Antibody | Anti-MDMX (mouse monoclonal) | Sigma-Aldrich | Cat# M0445; RRID:AB_532256 | (1:1000) |
| Antibody | Anti-p21 Waf1/Cip1 (12D1) (rabbit monoclonal) | Cell Signaling Technology | Cat# 2947; RRID:AB_823586 | (1:1000) |
| Antibody | Anti-α-actinin (HRP) (rabbit monoclonal) | Cell Signaling Technology | Cat# 12413; RRID:AB_2797903 | (1:1000) |
| Antibody | Anti-GAPDH (HRP) (rabbit monoclonal) | Cell Signaling Technology | Cat# 8884; RRID:AB_11129865 | (1:2000) |
| Antibody | Anti-nucleophosmin (mouse monoclonal) | Abcam | Cat# ab10530; RRID:AB_297271 | (1:500) |
| Antibody | Anti-Phospho-Histone H2A.X (Ser139) (20E3) (rabbit monoclonal) | Cell Signaling Technology | Cat# 9718; RRID:AB_2118009 | (1:250) |

*Appendix 1 Continued on next page*

*Appendix 1 Continued*

| Reagent type (species) or resource | Designation | Source or reference | Identifiers | Additional information |
|---|---|---|---|---|
| Antibody | Goat anti-mouse IgG (H+L) Cross-Adsorbed Secondary Antibody, Alexa Fluor 488 | Thermo Fisher Scientific | Cat# A-11001; RRID:AB_2534069 | (1:500) |
| Antibody | Goat anti-rabbit IgG (H+L) Highly Cross-Adsorbed Secondary Antibody, Alexa Fluor 594 | Thermo Fisher Scientific | Cat# A-11037; RRID:AB_2534095 | (1:500) |
| Antibody | Anti-GAPDH (HRP) (rabbit monoclonal) | Cell Signaling Technology | Cat# 8884; RRID:AB_11129865 | (1:2000) |
| Antibody | Goat anti-mouse IgG, Light chain specific (HRP) | Jackson ImmunoResearch Laboratories, Inc | Cat# 115-035-174; RRID:AB_2338512 | (1:5000) |
| Antibody | Goat anti-rabbit IgG, HRP-linked antibody | Cell Signaling Technology | Cat# 7074; RRID:AB_2099233 | (1:5000) |
| Recombinant DNA reagent | lentiCas9-Blast | PMID:25075903 | Addgene plasmid# 52962; RRID:Addgene_52962 | |
| Recombinant DNA reagent | psPAX2 | Addgene | Addgene plasmid# 12260; RRID:Addgene_12260 | |
| Recombinant DNA reagent | pMD2.G | Addgene | Addgene plasmid# 12259; RRID:Addgene_12259 | |
| Recombinant DNA reagent | Human GeCKOv2 CRISPR Knockout Pooled Library (A+B) in lentiGuide-PURO | PMID:25075903 | Addgene plasmid# 1000000048 | |
| Recombinant DNA reagent | lentiGuide-PURO | PMID:25075903 | Addgene plasmid# 52963; RRID:Addgene_52963 | |
| Commercial assay or kit | cOmplete, EDTA-free, Protease Inhibitor Cocktail | Roche | Cat# 11873580001 | |
| Commercial assay or kit | PhosSTOP | Roche | Cat# 4906837001 | |
| Commercial assay or kit | Pefabloc SC | Roche | Cat# 11429868001 | |
| Commercial assay or kit | TURBO DNase (2 U/µL) | Invitrogen | Cat# AM2238 | |
| Commercial assay or kit | RNase I, *E. coli* | Lucigen | Cat# N6901K | |
| Commercial assay or kit | SUPERaseIn RNase Inhibitor | Invitrogen | Cat# AM2694 | |
| Commercial assay or kit | TRIzol Reagent | Invitrogen | Cat# 15596018 | |
| Commercial assay or kit | SYBR Gold Nucleic Acid Gel Stain | Invitrogen | Cat# S11494 | |
| Commercial assay or kit | T4 Polynucleotide Kinase | New England BioLabs | Cat# M0201S | |
| Commercial assay or kit | T4 RNA Ligase 2, truncated K227Q | New England BioLabs | Cat# M0351S | |
| Commercial assay or kit | SuperScript III Reverse Transcriptase | Invitrogen | Cat# 18080085 | |
| Commercial assay or kit | Random Hexamers | Invitrogen | Cat# N8080127 | |

*Appendix 1 Continued*

| Reagent type (species) or resource | Designation | Source or reference | Identifiers | Additional information |
|---|---|---|---|---|
| Commercial assay or kit | CircLigase II ssDNA Ligase | Lucigen | Cat# CL9021K | |
| Commercial assay or kit | LD-Dithiothreitol | MilliporeSigma | Cat# D9779 | |
| Commercial assay or kit | Iodoacetamide | MilliporeSigma | Cat# I1149 | |
| Commercial assay or kit | o-Phosphoric acid, 85% | Fisher Scientific | Cat# A260-500 | |
| Commercial assay or kit | Water, Optima LC/MS Grade | Fisher Scientific | Cat# W6-4 | |
| Commercial assay or kit | Methanol, Optima LC/MS Grade | Fisher Scientific | Cat# A456 | |
| Commercial assay or kit | Triethylammonium bicarbonate buffer | MilliporeSigma | Cat# T7408 | |
| Commercial assay or kit | Trypsin Gold, Mass Spectrometry Grade | Promega | Cat# V5280 | |
| Commercial assay or kit | Formic Acid, LC/MS Grade | Thermo Scientific Pierce | Cat# 28905 | |
| Commercial assay or kit | Acetonitrile, Optima LC/MS Grade | Fisher Scientific | Cat# A955-1 | |
| Commercial assay or kit | Phusion High Fidelity DNA polymerase | New England BioLabs | Cat# M0530S | |
| Commercial assay or kit | BsmBIv2 | New England BioLabs | Cat# R0739S | |
| Commercial assay or kit | DNase I (RNase-free) | New England BioLabs | Cat# M0303S | |
| Commercial assay or kit | RNA Clean and Concentrator-25 | Zymo Research | Cat# R1017 | |
| Commercial assay or kit | RNasin Ribonuclease Inhibitor | Promega | Cat# N2515 | |
| Commercial assay or kit | SYBR Safe DNA Gel Stain | Invitrogen | Cat# S33102 | |
| Commercial assay or kit | Click-iT Cell Reaction Buffer Kit | Invitrogen | Cat# C10269 | |
| Commercial assay or kit | Neon Transfection System 10 µL Kit | Thermo Scientific | Cat# MPK1096 | |
| Commercial assay or kit | Gene Knockout Kit v2 – human – RPL22 | Synthego | NA | |
| Commercial assay or kit | Negative Control, Scrambled sgRNA#1, mod-sgRNA | Synthego | NA | |
| Commercial assay or kit | ProLong Gold Antifade Mountant | Thermo Fisher Scientific | Cat# P36934 | |
| Commercial assay or kit | Lipofectamine 3000 Transfection Reagent | Invitrogen | Cat# L3000075 | |
| Commercial assay or kit | QuantiGene Plex panel | Thermo Fisher Scientific | NA | |
| Commercial assay or kit | QuantiGene Sample Processing Kit for cultured cells | Thermo Fisher Scientific | Cat# QS0100 | |

*Appendix 1 Continued on next page*

*Appendix 1 Continued*

| Reagent type (species) or resource | Designation | Source or reference | Identifiers | Additional information |
|---|---|---|---|---|
| Commercial assay or kit | QuantiGene Plex Assay kits | Thermo Fisher Scientific | Cat# QP1013 | |
| Commercial assay or kit | Pierce BCA Protein Assay Kit | Thermo Scientific | Cat# PI23225 | |
| Commercial assay or kit | Clarity Western ECL Substrate | Bio-Rad | Cat# 1705061 | |
| Commercial assay or kit | Qubit RNA High Sensitivity Assay Kit | Invitrogen | Cat# Q32852 | |
| Commercial assay or kit | Direct-zol RNA Miniprep | Zymo Research | Cat# R2050 | |
| Commercial assay or kit | RiboCop rRNA Depletion Kit V1.2 | Lexogen | Cat# 037.24 | |
| Commercial assay or kit | Quick-DNA MidiPrep Plus Kit | Zymo Research | Cat# D4075 | |
| Commercial assay or kit | NEBNext High Fidelity 2X PCR Master Mix | New England BioLabs | Cat# M0541L | |
| Commercial assay or kit | Zymo-Spin V Columns with Reservoir | Zymo Research | Cat# C1016-25 | |
| Commercial assay or kit | NucleoSpin Gel and PCR Clean-up | Macherey-Nagel | Cat# 740609.250 | |
| Commercial assay or kit | Gibson Assembly Master Mix | New England BioLabs | Cat# E2611S | |
| Commercial assay or kit | NucleoBond Xtra Maxi EF | Macherey-Nagel | Cat# 740424.10 | |
| Commercial assay or kit | Q5 DNA Polymerase | New England BioLabs | Cat# M0491S | |
| Commercial assay or kit | Taq DNA Polymerase with Standard Taq Buffer | New England BioLabs | Cat# M0273S | |
| Commercial assay or kit | SYBR Safe DNA Gel Stain | Invitrogen | Cat# S33102 | |
| Commercial assay or kit | KAPA SYBR Fast qPCR Master Mix (2×) | Roche | Cat# 07959397001 | |
| Commercial assay or kit | CellTiter-Glo Luminescent Cell Viability Assay | Promega | Cat# G7572 | |
| Chemical compound, drug | Blasticidin S Hydrochloride Powder | Research Products International | Cat# B12200-0.05 | |
| Chemical compound, drug | DMSO | Sigma | Cat# D2650 | |
| Chemical compound, drug | C6 | PMID:30865883 | N/A | |
| Chemical compound, drug | C16 | PMID:31858797 | N/A | |
| Chemical compound, drug | 2'-Azido-2'-deoxycytidine | Biosynth | Cat# NA05412 | |
| Chemical compound, drug | Actinomycin D | Cayman Chemical Company | Cat# 11421-10mg | |
| Chemical compound, drug | MB 680R DBCO | Vector Laboratories | Cat# CCT-1462 | |

*Appendix 1 Continued on next page*

*Appendix 1 Continued*

| Reagent type (species) or resource | Designation | Source or reference | Identifiers | Additional information |
|---|---|---|---|---|
| Chemical compound, drug | Cycloheximide | Research Products International | Cat# C81040-1.0 | |
| Chemical compound, drug | OPP (*O*-propargyl-puromycin) | Invitrogen | Cat# C10459 | |
| Chemical compound, drug | Hoechst 33342 | Thermo Fisher Scientific | Cat# H3570 | |
| Chemical compound, drug | Alexa Fluor 647 Azide, Triethylammonium Salt | Invitrogen | Cat# A10277 | |
| Chemical compound, drug | Cycloheximide | Sigma | Cat# C4859-1ML | |
| Chemical compound, drug | Nutlin-3a | Cayman Chemical Company | Cat# 18585 | |
| Chemical compound, drug | Rapamycin | MedChem Express | Cat# HY-10219 | |
| Chemical compound, drug | Pinometostat | Cayman Chemical Company | Cat# 16175 | |
| Chemical compound, drug | Harmine | Sigma-Aldrich | Cat# 286044 | |
| Chemical compound, drug | Mivebresib | Cayman Chemical Company | Cat# 21033 | |
| Chemical compound, drug | Venetoclax | Cayman Chemical Company | Cat# 16233 | |
| Chemical compound, drug | Etoposide | Cayman Chemical Company | Cat# 12092 | |
| Chemical compound, drug | Olaparib | Cayman Chemical Company | Cat# 10621 | |
| Chemical compound, drug | VE-821 | Cayman Chemical Company | Cat# 17587 | |
| Chemical compound, drug | Pemrametostat | Selleck Chemicals | Cat# S8664 | |
| Chemical compound, drug | Alvespimycin | Cayman Chemical Company | Cat# 11036 | |
| Software, algorithm | R | The R Foundation | RRID:SCR_001905 | https://www.r-project.org |
| Software, algorithm | drc | *Ritz et al., 2015*; *Ritz and Streibig, 2021* | NA | https://github.com/DoseResponse/drc |
| Software, algorithm | CHOPCHOP | *Labun et al., 2019* | RRID:SCR_015723 | https://chopchop.cbu.uib.no |
| Software, algorithm | count_spacers.py | *Joung et al., 2017*; *Joung, 2017* | NA | https://github.com/fengzhanglab/Screening_Protocols_manuscript/blob/master/design_targeted_library.py |
| Software, algorithm | SynergyFinder Plus | *Zheng et al., 2022* | RRID:SCR_019318 | https://synergyfinder.org/ |
| Software, algorithm | PASWR | *Ugarte et al., 2015*; *Arnholt, 2022* | NA | https://github.com/cran/PASWR |
| Software, algorithm | cutadapt | *Martin, 2011* | RRID:SCR_011841 | https://github.com/marcelm/cutadapt/ |

*Appendix 1 Continued on next page*

*Appendix 1 Continued*

| Reagent type (species) or resource | Designation | Source or reference | Identifiers | Additional information |
|---|---|---|---|---|
| Software, algorithm | sabre | NA | RRID:SCR_011843 | https://github.com/najoshi/sabre |
| Software, algorithm | bowtie2 | *Langmead and Salzberg, 2012* | RRID:SCR_016368 | https://github.com/BenLangmead/bowtie2 |
| Software, algorithm | STAR | *Dobin et al., 2013* | RRID:SCR_004463 | https://github.com/alexdobin/STAR |
| Software, algorithm | UMI-tools | *Smith et al., 2017* | RRID:SCR_017048 | https://github.com/CGATOxford/UMI-tools |
| Software, algorithm | BEDTools | *Quinlan and Hall, 2010* | RRID:SCR_006646 | https://github.com/arq5x/bedtools2 |
| Software, algorithm | Xtail | *Xiao et al., 2016; xryanglab, 2016* | NA | https://github.com/xryanglab/xtail |
| Software, algorithm | riboWaltz | *Lauria et al., 2018* | RRID:SCR_016948 | https://github.com/LabTranslationalArchitectomics/riboWaltz |
| Software, algorithm | featureCounts | *Liao et al., 2014* | RRID:SCR_012919 | http://bioconductor.org/packages/release/bioc/html/Rsubread.html |
| Software, algorithm | DESeq2 | *Love et al., 2014* | RRID:SCR_015687 | https://bioconductor.org/packages/release/bioc/html/DESeq2.html |
| Software, algorithm | MAGeCK | *Li et al., 2014; Li and Song, 2022* | NA | https://sourceforge.net/p/mageck/wiki/Home/ |
| Software, algorithm | Maxquant | *Cox and Mann, 2008* | RRID:SCR_014485 | https://cox-labs.github.io/coxdocs/maxquant_instructions.html |
| Software, algorithm | Andromeda | *Cox et al., 2011* | NA | https://cox-labs.github.io/coxdocs/andromeda_instructions.html |
| Software, algorithm | Msstats | *Choi et al., 2014* | RRID:SCR_014353 | https://msstats.org/ |
| Software, algorithm | fgsea | *Korotkevich et al., 2021* | RRID:SCR_020938 | https://bioconductor.org/packages/release/bioc/html/fgsea.html |
| Software, algorithm | Biorender | NA | RRID:SCR_018361 | https://biorender.com |
| Software, algorithm | PyMOL | NA | RRID:SCR_000305 | https://pymol.org/2/ |
| Software, algorithm | rMATS | *Shen et al., 2014* | RRID:SCR_013049 | https://rnaseq-mats.sourceforge.net |
| Software, algorithm | Molecular Signatures Database | *Liberzon et al., 2011* | RRID:SCR_016863 | https://www.gsea-msigdb.org/gsea/msigdb/index.jsp |
| Software, algorithm | Universal Protein Resource | *Bateman et al., 2021* | RRID:SCR_002380 | https://www.uniprot.org |
| Software, algorithm | FlowJo | NA | RRID:SRC_008520 | https://www.flowjo.com/solutions/flowjo |
| Software, algorithm | Empiria Studio | LI-COR | RRID:SCR_022512 | https://www.licor.com/bio/empiria-studio/ |
| Software, algorithm | NIS-Elements AR 5.42.03 64-bit | Nikon Instruments | RRID:SCR_014329 | https://www.nikoninstruments.com/Products/Software |

*Appendix 1 Continued on next page*

*Appendix 1 Continued*

| Reagent type (species) or resource | Designation | Source or reference | Identifiers | Additional information |
|---|---|---|---|---|
| Software, algorithm | Fiji | Fiji/ImageJ | RRID:SCR_002285 | http://fiji.sc |
| Other | S-Trap Micro Columns | ProtiFi | Cat# C02-micro-80 | |
| Other | Jupiter 3 um C18 300A, Bulk packaging | Phenomenex | Cat# 04A-4263 | |
| Other | Molex Polymicro Capillary 100 um × 363 um | Fisher Scientific | Cat# 50-110-8623 | |
| Other | Neon Transfection System | Thermo Fisher Scientific | Cat# MPK5000 | |
| Other | Luminex FLEXMAP 3D System | Invitrogen | Cat# APX1342 | |
| Other | 4–20% Mini-PROTEAN TGX Precast Gel | Bio-Rad | Cat# 4561096 | |
| Other | Amersham Protran Western Blotting Membranes, Nitrocellulose | Cytiva | Cat# GE10600001 | |
| Other | C1000 Touch Thermal Cycler Chassis | Bio-Rad | Cat# 1841100 | |
| Other | CFX96 Optical Reaction Module for Real-Time PCR System | Bio-Rad | Cat# 1845097 | |
| Other | ChemiDoc Imaging System | Bio-Rad | Cat# 17001401 | |
| Other | Thick-wall Polycarbonate Tubes, 13 × 51 mm | Beckman-Coulter | Cat# 349622 | |
| Other | Novex TBE-Urea Gels 15%, 12 well | Invitrogen | Cat# EC68852BOX | |
| Other | Costar Spin-X Centrifuge Tube Filters | Corning | Cat# CLS8162 | |
| Other | Novex TBE Gels, 8%, 15 well | Invitrogen | Cat# EC62155BOX | |
| Other | GloMax Explorer Multimode Microplate Reader | Promega | Cat# GM3500 | |
| Other | Orbitrap Exploris 480 Mass Spectrometer | Thermo Scientific | Cat# BRE725533 | |

