## [Editor Report · eLife assessment]

This **important** article reveals that one of the major roles of the WDR5 WIN site is to promote ribosome synthesis, and that by attacking the WIN site with inhibitors ribosome attrition occurs creating new vulnerabilities that can be therapeutically exploited. This deficiency of ribosomal proteins also provokes the p53 response. The data from a variety of approaches is generally very **convincing**, and together buttresses the authors' conclusions and interpretations quite nicely; overall, this article will provide a justification for pre-clinical and translational studies of WDR5 interaction site inhibitors.

---

## [Referee Report · Reviewer #1 (Public Review)]

Building on previous work from the Tansey lab, here Howard et al. characterize transcriptional and translational changes upon WIN site inhibition of WDR5 in MLL-rearranged cancer cells. They first analyze whether C16, a newer generation compound, has the same cellular effects as C6, an early generation compound. Both compounds reduce the expression of WDR5-bound RPGs in addition to the unbound RPG RPL22L1. They then investigate differential translation by ribo-seq and observe that WIN site inhibition reduces the translational RPGs and other proteins related to biomass accumulation (spliceosome, proteasome, mitochondrial ribosome). Interestingly, this reduction adds to the transcriptional changes and is not limited to RPGs whose promoters are bound by WDR5. Quantitative proteomics at two time points confirmed the downregulation of RPGs. Interestingly, the overall effects are modest, but RPL22LA is strongly affected. Unexpectedly, most differentially abundant proteins seem to be upregulated 24 h after C6 (see below). A genetic screen showed that loss of p53 rescues the effect of C6 and C16 and helped the authors to identify pathways that can be targeted by compounds together with WIN site inhibitors in a synergistic way. Finally, the authors elucidated the underlying mechanisms and analyzed the functional relevance of the RPL22, RPL22L1, p53 and MDM4 axis.

Comments on revised version:

The authors have answered my points satisfactorily and the manuscript has become clearer and more meaningful as a result. In particular, the measurement of global translation rate is important and validates the upregulation of a number of proteins following WDR5 inhibitor treatment.

---

## [Referee Report · Reviewer #2 (Public review)]

Summary:

The manuscript by Howard et al reports the development of high affinity WDR5-interaction site inhibitors (WINi) that engage the protein to block the arginine-dependent engagement with its partners. Treatment of MLL-rearranged leukemia cells with high-affinity WINi (C16) decreases the expression of genes encoding most ribosomal proteins and other proteins required for translation. Notably, although these targets are enriched for WDR5-ChIP-seq peaks, such peaks are not universally present in the target genes. High concordance was founded between the alterations in gene expression due to C16 treatment and the changes resulting from treatment with an earlier, lower affinity WINi (C6). Besides protein synthesis, genes involved in DNA replication or MYC responses are downregulated, while p53 targets and apoptosis genes are upregulated. Ribosome profiling reveals a global decrease in translational efficiency due to WINi with overall ribosome occupancies of mRNAs ~50% of control samples. The magnitude in the decrements of translation for most individual mRNAs exceeds the respective changes in mRNA levels genome-wide. From these results and other considerations, the authors hypothesize that WINi results in ribosome depletion. Quantitative mass spec documents the decrement in ribosomal proteins following WINi treatment along with increases in p53 targets and proteins involved in apoptosis occurring over 3 days. Notably RPL22L1 is essentially completely lost upon WINi treatment. The investigators next conduct a CRISPR screen to find moderators and cooperators with WINi. They identify components of p53 and DNA repair pathways as mediators of WINi inflicted cell death (so gRNAs against these genes permit cell survival). Next, WINi are tested in combination with a variety of other agents to explore synergistic killing to improve their expected therapeutic efficacy. The authors document loss of the p53 antagonist MDM4 (in combination with splicing alterations of RPL22L1), an observation that supports the notion that WINi killing is p53-mediated.

This is a scientifically very strong and well-written manuscript that applies a variety of state-of-the art molecular approaches to interrogate the role of the WDR5 interaction site and WINi. They reveal that the effects of WINi seem to be focused on the overall synthesis of protein components of the translation apparatus, especially ribosomal proteins-even those that do not bind WDR5 by ChIP (a question left unanswered is how such the WDR5-less genes are nevertheless WINi targeted). They convincingly show that disruption of the synthesis of these proteins occurs upon activation of p53 dependent apoptosis, likely driven by unbalanced ribosomal protein synthesis leading to MDM2 inhibition. This apoptosis is subsequently followed, as expected by ɣH2AX-activation. Pathways of possible WINi resistance and synergies with other anti-neoplastic approaches are explored. These experiments are all well-executed and strongly invite more extensive pre-clinical and translational studies of WINi in animal studies. The studies also may anticipate the use of WINi as probes of nucleolar function and ribosome synthesis though this was not really explored in the current manuscript. The current version of the manuscript documents ribosomal stress revealed by leakage of NPM1 into the nucleoplasm while nucleolar integrity is preserved. A progressive loss of rRNA synthesis occurs upon drug treatment that is presumably secondary to the decrement in ribosomal protein production.

Comments on revised version:

(1) The authors to my mind, have quite nicely and professionally addressed the comments of the reviewers and are to be congratulated on an important contribution to the elucidation of WDR5 biology and pathology.

---

## [Author Response]

The following is the authors’ response to the original reviews.

We are grateful to the Editors for overseeing the review of our manuscript, and to the two reviewers for their thoughtful comments and suggestions for how it can be improved.

I submit at this time a revision, as well as a detailed response (below) to each of the points raised in the first round of review.

We feel the manuscript has been significantly improved by taking the reviewers' comments to heart. In a nutshell, we added new key pieces of data (impact of WIN site inhibition on global translation, rRNA production, as well as the requested cell biology analyses showing nucleolar stress), new analyses of the proteomics to counter potential concerns with normalization, and expanded/revised verbiage in key areas to clarify parts of the text that were confusing or problematic. The main figures have not changed; all new material is included in supplements to figures 2 and 3.

**Public Reviews**

**Reviewer #1 (Public Review):**
Building on previous work from the Tansey lab, here Howard et al. characterize transcriptional and translational changes upon WIN site inhibition of WDR5 in MLL-rearranged cancer cells. They first analyze whether C16, a newer generation compound, has the same cellular effects as C6, an early generation compound. Both compounds reduce the expression of WDR5-bound RPGs in addition to the unbound RPG RPL22L1. They then investigate differential translation by ribo-seq and observe that WIN site inhibition reduces the translational RPGs and other proteins related to biomass accumulation (spliceosome, proteasome, mitochondrial ribosome). Interestingly, this reduction adds to the transcriptional changes and is not limited to RPGs whose promoters are bound by WDR5. Quantitative proteomics at two-time points confirmed the downregulation of RPGs. Interestingly, the overall effects are modest, but RPL22LA is strongly affected. Unexpectedly, most differentially abundant proteins seem to be upregulated 24 h after C6 (see below). A genetic screen showed that loss of p53 rescues the effect of C6 and C16 and helped the authors to identify pathways that can be targeted by compounds together with WIN site inhibitors in a synergistic way. Finally, the authors elucidated the underlying mechanisms and analyzed the functional relevance of the RPL22, RPL22L1, p53, and MDM4 axis.While this work is not conceptually new, it is an important extension of the observations of Aho et al. The results are clearly described and, in my view, very meaningful overall.Major points:(1) The authors make statements about the globality/selectivity of the responses in RNA-seq, ribo-seq, and quantitative proteomics. However, as far as I can see, none of these analyses have spike-in controls. I recommend either repeating the experiments with a spike-in control or carefully measuring transcription and translation rates upon WIN site inhibition and normalizing the omics experiments with this factor.

The reviewer is correct that we did not include spike-in controls in our omics experiments. We would like to emphasize that none of the omics data in this manuscript have been processed in unorthodox ways, and that the major conclusions each have independent corroborating data.

The selectivity in RPG suppression observed in RNA-Seq, for example, is supported by results from our target engagement (QuantiGene) assays; suppression of RPL22L1 mRNA levels is supported by quantitative and semi-quantitative RT-PCR, by western blotting, and by the results of our proteomic profiling; alternative splicing (and expression) of MDM4—and its dependency on RPL22—is also backed up by similar RT-PCR and western blotting data. The same applies for alternative splicing of RPL22L1.

That said, we do appreciate the point the reviewer is making here, and have done our best to respond. We do not think it is a prudent investment in resources to repeat the numerous omics assays in the manuscript. We also considered normalizing for bulk transcription and translation rates as suggested, but it is not clear in practice how this would be done, and it could introduce additional variables and uncertainties that may skew the interpretation of results. Instead, to respond to this comment, we made the following changes to the manuscript:

(1) We now explicitly state, for all omics assays, that spike-in controls were not included. These statements will prompt the reader to make their own assessment of the robustness of each of our findings and interpretations.

(2) We have added new data to the manuscript (Figure 2—figure supplement 1A–B) measuring the impact of C6 and C16 on bulk translation using the OPP labeling method. These new data demonstrate that WIN site inhibitors induce a progressive yet modest decline in protein synthesis capacity. At 24 hours, there is no significant effect of either agent on protein synthesis levels. By 48 hours, a small but significant effect is observed, and by 96 hours translation levels are ~60% of what they are in vehicle-treated control cells. These new data are important because they support the idea that normalization has not blunted the responses we observe—the magnitude of the effects are consistent between the different assays and tend to cap out at two-fold in terms of RPG suppression, translation efficiency, ribosomal protein levels, and protein synthesis capacity.

(3) We have included additional analysis regarding the LFQMS, as described below, that specifically addresses the issue of normalization in our proteomics experiments.

(2) Why are the majority of proteins upregulated in the proteomics experiment after 24 h in C6 (if really true after normalization with general protein amount per cell)? This is surprising and needs further explanation.

The reviewer is correct in noting that (by LFQMS) ~700 proteins are induced after 24 hours of treatment of MV4:11 cells with C16 (not C6, as stated). The reviewer would like us to examine whether this apparent increase in proteins is a normalization artifact. In response to this comment, we have made the following changes to the manuscript:

(1) Our new OPP labeling experiments (Figure 2—figure supplement 1A–B) show that there is no significant reduction in overall protein synthesis following 24 hours of C16 treatment. In light of this finding, it is unlikely that normalization artifacts, resulting from diminution of the pool of highly abundant proteins, create the appearance of these 700 proteins being induced. We now explicitly make this point in the text.

(2) We now clarify in the methods how we seeded identical numbers of cells for DMSO and C16-treated cultures in these experiments, and—consistent with our finding that WIN site inhibitors have little if any effect on protein synthesis or proliferation at the 24 hour timepoint— extracted comparable amounts of proteins from these two treatment conditions (DMSO: 344.75 ± 21.7 µg; C16: 366.50 ± 15.8 µg; [Mean ± SEM]).

(3) We now include in Figure 3—figure supplement 1A a plot showing the distribution of peptide intensities for each protein detected in each run of LFQMS before and after equal median normalization. This new analysis reveals that the distribution of intensities is not appreciably changed via normalization. Specifically, there is not a reduction in peptide intensities in the unnormalized data from 24 hours of C16 treatment that is reversed or tempered by normalization. This analysis provides further support for the notion that the increase we observe is not a normalization artifact.

(4) We now include in Figure 3—figure supplement 1B–D a set of new analyses examining the relationship between the initial intensity of proteins in DMSO control samples (a crude proxy for abundance) versus the fold change in response to WIN site inhibitor. This analysis shows that we have as many "highly abundant" (10th decile) proteins increasing as we do decreasing in response to WINi. Thus, it appears as though the wholesale clearance of highly abundant proteins from the cell is not occurring at this early treatment timepoint. In addition, this analysis also shows that ribosomal proteins (RP) are generally the most abundant, most suppressed, proteins and that their fold-change at the protein level at 24 hours is less than two-fold, consistent again with the magnitude of transcriptional effects of C16, as measured by RNA-Seq and QuantiGene. The fact that the drop in RP levels is consistent with expectations based on other analyses provides further empirical support for the notion that protein levels inferred from LFQMS are authentic and not skewed by global changes in the proteome.

The increase in proteins at this time point, we argue, is thus most likely genuine. It is not surprising that—at a timepoint at which protein synthesis is unaffected—several hundred proteins are induced by a factor of two. How this occurs, we do not know. It may be a transient compensatory mechanism, or it may be an early part of the active response to WIN site inhibitors. Lest the reader be confused by this finding, we have now added text to this section of the manuscript discussing and explaining the phenomenon in more detail.

(3) The description of the two CRISPR screens (GECKO and targeted) is a bit confusing. Do I understand correctly that in the GECKO screen, the treated cells are not compared with nontreated cells of the same time point, but with a time point 0? If so, this screen is not very meaningful and perhaps should be omitted. Also, it is unclear to me what the advantages of the targeted screen are since the targets were not covered with more sgRNAs (data contradictory: 4 or 10 sgRNAs per target?) than in Gecko. Also, genome-wide screens are feasible in culture for multiple conditions. Overall, I find the presentation of the screening results not favorable.

In essence, this is a single screen performed in two tiers. In Tier 1, we screened a complete GECKO library (six sgRNA/gene) with the earliest generation (less potent) inhibitor C6, and compared sgRNA representation against the time zero population. This screen would reveal sgRNAs that are specifically associated with response to C6, as well as those that are associated with general cell fitness and viability. We then identified genes connected to these sgRNAs, removed those that are pan essential, and built a custom library for the second tier using sgRNAs from the Brunello library (four sgRNA/gene). We then screened this custom library with both C6 and the more potent inhibitor C16, this time against DMSO-treated cells from the same timepoint.

We acknowledge that this is not the most streamlined setup for a screen. But our intention was to compare two inhibitors (C6 and C16) and identify high confidence 'hits' that are disconnected from general cell viability, rather than generate an exhaustive list of all genes that, when disrupted, skew the response to WIN site inhibitor. The final result of this screen (Figure 4E) is a gene list that has been validated with two chemically distinct WIN site inhibitors and up to 10 unique sgRNAs per gene. We may not have captured every gene that can modulate response to WIN site inhibitor, but those appearing in Figure 4E are highly validated.

To answer the reviewer's specific questions: (i) we cannot omit the Tier 1 screen because then there would be no rationale for what was screened in the second Tier; and (ii) the advantage of the custom Tier 2 library is that it allowed us to screen hits from the Tier 1 screen with four completely independent sgRNAs. Although there are not more sgRNAs for each gene in the Tier 2 versus the Tier 1 library, these sgRNAs are different and thus, for C6 at least, hits surviving both screens were validated with up to 10 unique sgRNAs.

We apologize that the description of the CRISPR screens was not clearer, and have reworked this section of the manuscript to make our intent and our actions clearer.

(4) Can Re-expression of RPL22 rescue the growth arrest of C6?.

We have not attempted to complement the RPL22 knock out. But we do note that evidence supporting the idea that loss of RPL22 confers resistance to WIN site inhibitor is strong—six (out of six) sgRNAs against RPL22 were significantly enriched in the Tier 1 screen, and independent knock out of RPL22 with the Synthego multi-guide system in MV4;11 and MOLM13 cells increases the GI50 for C16.

**Reviewer #2 (Public Review):**
Summary:The manuscript by Howard et al reports the development of high-affinity WDR5-interaction site inhibitors (WINi) that engage the protein to block the arginine-dependent engagement with its partners. Treatment of MLL-rearranged leukemia cells with high-affinity WINi (C16) decreases the expression of genes encoding most ribosomal proteins and other proteins required for translation. Notably, although these targets are enriched for WDR5-ChIP-seq peaks, such peaks are not universally present in the target genes. High concordance was found between the alterations in gene expression due to C16 treatment and the changes resulting from treatment with an earlier, lower affinity WINi (C6). Besides protein synthesis, genes involved in DNA replication or MYC responses are downregulated, while p53 targets and apoptosis genes are upregulated. Ribosome profiling reveals a global decrease in translational efficiency due to WINi with overall ribosome occupancies of mRNAs ~50% of control samples. The magnitude of the decrements of translation for most individual mRNAs exceeds the respective changes in mRNA levels genome-wide. From these results and other considerations, the authors hypothesize that WINi results in ribosome depletion. Quantitative mass spec documents the decrement in ribosomal proteins following WINi treatment along with increases in p53 targets and proteins involved in apoptosis occurring over 3 days. Notably, RPL22L1 is essentially completely lost upon WINi treatment. The investigators next conduct a CRISPR screen to find moderators and cooperators with WINi. They identify components of p53 and DNA repair pathways as mediators of WINi-inflicted cell death (so gRNAs against these genes permit cell survival). Next, WINi are tested in combination with a variety of other agents to explore synergistic killing to improve their expected therapeutic efficacy. The authors document the loss of the p53 antagonist MDM4 (in combination with splicing alterations of RPL22L1), an observation that supports the notion that WINi killing is p53-mediated.Strengths:This is a scientifically very strong and well-written manuscript that applies a variety of state-ofthe art molecular approaches to interrogate the role of the WDR5 interaction site and WINi. They reveal that the effects of WINi seem to be focused on the overall synthesis of protein components of the translation apparatus, especially ribosomal proteins-even those that do not bind WDR5 by ChIP (a question left unanswered is how much the WDR5-less genes are nevertheless WINi targeted). They convincingly show that disruption of the synthesis of these proteins is accompanied by DNA damage inferred by H2AX-activation, activation of the p53pathway, and apoptosis. Pathways of possible WINi resistance and synergies with other antineoplastic approaches are explored. These experiments are all well-executed and strongly invite more extensive pre-clinical and translational studies of WINi in animal studies. The studies also may anticipate the use of WINi as probes of nucleolar function and ribosome synthesis though this was not really explored in the current manuscript.Weaknesses:A mild deficiency in the current manuscript is the absence of cell biological methods to complement the molecular biological and biochemical approaches so ably employed. Some microscopic observations and confirmation of nucleolar dysfunction and DNA damage would be reassuring.

We thank the reviewer for their comments. We agree that an absence of cell biological methods was a deficiency in the original manuscript. In response to this comment, we have now added immunofluorescence (IF) analyses, examining the impact of C16 on nucleolar integrity and nucleophosmin (NPM1) distribution (Figure 3—figure supplement 4). These new data clearly show that C16 induces nucleolar stress at 72 hours—as measured by the redistribution of NPM1 from the nucleolus to the nucleoplasm. These new data fill an important gap in the story, and we are grateful to the reviewer for prompting us to perform these experiments.

As part of the above study, we also probed for gamma-H2AX, expecting that we may see some signs of accumulation in the nucleoli (see comment #4 from Reviewer #2, below). We did not observe this response. Importantly, however, we did see that gamma-H2AX staining occurs only in what are overtly apoptotic cells. This is an important finding, because we had previously speculated that the induction of gamma-H2AX observed by Western blotting reflected part of a bona-fide response to DNA damage elicited by WIN site inhibitors. Instead, the IF data now leads us to conclude that this signal simply reflects the established fact that WIN site inhibitors induce apoptosis in this cell line (Aho et al., 2019). In response to this new finding, we have added additional discussion to the text and have removed or de-emphasized the potential contribution of DNA damage to the mechanism of action of WDR5 WIN site inhibitors. Again, we are grateful for this comment as it has prevented us from continuing to report/pursue erroneous observations.

**Recommendations for the authors**

**Reviewer #1 (Recommendations For The Authors):**
There is a typo in "but are are linked to mRNA instability when translation is inhibited".

Thank you for catching this typo. It has now been corrected.

**Reviewer #2 (Recommendations For The Authors):**
(1) The authors report that WINi initially (at 24 hrs) increases the expression of most proteins while decreasing ribosomal proteins, but at 72 hours all proteins are depressed. The transient bump-up of non-translation-related proteins seems odd. A simple resolution to this somewhat strange observation is that there is no real increase in the other proteins, but because of the loss of a large fraction of the most abundant cellular proteins (the ribosomal proteins), the relative fraction of all other proteins is increased; that is, the increase of non-ribosomal proteins may be an artifact of normalization to a lower total protein content. Can this be explored?

We are grateful to the reviewer for this comment. We have tried our best to respond, as detailed above in response to Reviewer #1 Public Comment #2.

(2) It would be really nice to assess nucleolar status microscopically. Do nucleoli get bigger? Smaller? Do they have abnormal morphology? Is there nucleolar stress? What happens to rRNA synthesis and processing?

We agree and thank the reviewer for raising this point. As noted in our response to Reviewer #2, above, we have included new IF that shows: (i) no obvious effect on nucleolar integrity, (ii) redistribution of NPM1 to the nucleoplasm (indicative of nucleolar stress), and (iii) induction of gamma-H2AX staining in apoptotic cells (indicative of apoptosis).

Additionally, in response to this comment, we also looked at the impact of WIN site inhibitors on rRNA synthesis, using AzCyd labeling. These new data appear in Figure 3—figure supplement 3. Interestingly, these new data show that there is a progressive decline in rRNA synthesis, and that by 96 hours of treatment levels of both 18S and 28S rRNAs are reduced— again by about a factor of two. Our interpretation of this finding is that in response to the progressive decline in RPG transcription there is a secondary decrease in rRNA synthesis. This result is perhaps not surprising, but it does again add an important missing piece to our characterization of WIN site inhibitors and is further support for the concept that inhibition of ribosome production is a dominant part of the response to these agents.

(3) The WINi elicited DNA damage is incompletely characterized, rather it is inferred from H2AX activation. Comet assays would help to confirm such damage.

As noted in our response to Reviewer #2, our original inference of DNA damage, prompted by gamma-H2AX activation, is erroneous, and due instead to the ability of WIN site inhibitors to induce apoptosis. We thus did not pursue comet assays, etc., and removed discussion of potential DNA damage from the manuscript.

(4) Staining and microscopic observation of H2AX would be very useful. Is the WINi provoked DNA damage nucleolar-localized? Does the deficiency of ribosomal proteins lead to localized genotoxic nucleolar stress - or alternatively does the paucity of ribosomes and decreased translation lead to imbalances in other cellular pathways, perhaps including some involved in overall genome maintenance which would provoke more global DNA damage and H2AX staining, not limited to the nucleolus.

Again, please see our response to the Public Comment from Reviewer #2.

(5) It would be important to assess the influence and effects of WINi on some p53 mutant, p53-/- and p53 wild-type cell lines. Given their prevalence, p53 status may be expected to alter WINi efficacy.

The issue of how p53 status impacts the response to WINi is interesting and important, but we feel this is beyond the scope of the current manuscript. It is likely that many factors contribute to the response of cancer cells to these agents, and thus simply surveying some cancer lines for their response and linking this to their p53 status is unlikely to be very informative. Making definitive statements about the contribution of p53, and the differences between wild-type, lossof-function mutants, gain of function mutants, and null mutants will require more extensive analyses and is fertile territory for future studies, in our opinion.